# Explainable artificial intelligence to identify follicles that optimize clinical outcomes during assisted conception

Simon Hanassab[1,2,3], Scott M. Nelson[4,5], Artur Akbarov[2], Arthur C. Yeung[1,6], Artsiom Hramyka[7], Toulin Alhamwi[1], Rehan Salim[6], Alexander N. Comninos [1,6], Geoffrey H. Trew[1,5], Tom W. Kelsey [7], Thomas Heinis[2,8], Waljit S. Dhillo [1,6,8] & Ali Abbara [1,6,8] ✉

Infertility affects one-in-six couples, often necessitating in vitro fertilization treatment (IVF). IVF generates complex data, which can challenge the utilization of the full richness of data during decision-making, leading to reliance on simple 'rules-of-thumb'. Machine learning techniques are well-suited to analyzing complex data to provide data-driven recommendations to improve decision-making. In this multi-center study ($n = 19,082$ treatment-naive female patients), including 11 European IVF centers, we harnessed explainable artificial intelligence to identify follicle sizes that contribute most to relevant downstream clinical outcomes. We found that intermediately-sized follicles were most important to the number of mature oocytes subsequently retrieved. Maximizing this proportion of follicles by the end of ovarian stimulation was associated with improved live birth rates. Our data suggests that larger mean follicle sizes, especially those >18 mm, were associated with premature progesterone elevation by the end of ovarian stimulation and a negative impact on live birth rates with fresh embryo transfer. These data highlight the potential of computer technologies to aid in the personalization of IVF to optimize clinical outcomes pending future prospective validation.

Infertility affects one in six couples worldwide and poses a significant challenge to population health, being recognized by the World Health Organization as one of the most serious global disabilities[1]. Assisted reproductive technology (ART), including in vitro fertilization (IVF) treatment, has emerged as a valuable intervention to help patients suffering from infertility, with the number of cycles conducted increasing annually[1]. IVF protocols are designed for the typical patient, with clinicians using their experience and expertize to personalize treatment for each individual. Moreover, IVF treatment cycles generate a large amount of complex data, which can be challenging for clinicians to fully assimilate and utilize when making such treatment decisions. This can result in the common practice of using simplified "rules-of-thumb" to ease interpretation. This approach, however, can be reductionist and result in the under-utilization of the rich information available. In this scenario, explainable artificial intelligence (XAI), and more specifically machine learning (ML) techniques, with their ability to handle complex large datasets, present an opportunity to enhance the personalization and efficacy of ART treatments[2].

[1]Department of Metabolism, Digestion, and Reproduction, Imperial College London, London, UK. [2]Department of Computing, Imperial College London, London, UK. [3]UKRI Centre for Doctoral Training in AI for Healthcare, Imperial College London, London, UK. [4]School of Medicine, University of Glasgow, Glasgow, UK. [5]TFP Fertility, Institute of Reproductive Sciences, Oxford, UK. [6]Imperial College Healthcare NHS Trust, London, UK. [7]School of Computer Science, University of St Andrews, St Andrews, UK. [8]These authors jointly supervised this work: Thomas Heinis, Waljit S. Dhillo and Ali Abbara. ✉e-mail: ali.abbara@imperial.ac.uk

A key decision in IVF treatment is to determine when to initiate the next stage of IVF treatment after ovarian stimulation (OS), namely the "trigger of oocyte maturation". The trigger is typically human chorionic gonadotropin (hCG) or a gonadotropin-releasing hormone (GnRH) agonist that provides luteinizing hormone (LH)-like exposure to enable oocytes to recommence meiosis and become "mature", thus attaining competence for fertilization by sperm[3]. This is a key step in IVF treatment protocols and impacts the number of mature oocytes retrieved and the success of treatment. However, ovarian follicles at the time of trigger administration that are too small, or too large (termed "post-mature"), are less likely to yield oocytes[4–6]. Therefore, the size of ovarian follicles is monitored during OS to determine the optimal time to administer the trigger.

The number of oocytes is positively associated with cumulative live birth rates especially at lower oocyte numbers, such that optimizing the oocyte yield is a clinically relevant treatment aim[7,8]. Indeed, understanding the determinants of oocyte yield has been identified by an international research priority-setting consortium[9], but only a few studies to date have aimed to determine the specific follicle sizes during OS that are most likely to yield mature oocytes, which in turn can inform when to complete the OS phase of IVF protocols[5,10,11].

A recent prospective study (n = 157 patients) aimed to determine the follicle sizes on the day of oocyte pickup (OPU) (which typically occurs 36–38 h after the trigger) that were most likely to yield mature oocytes and reaffirmed that smaller follicle sizes are less likely[5]. However, the decision of when to administer the trigger is taken during OS (typically ~2 days prior to the day of OPU), challenging retrospective extrapolation of these data. Similarly, taking into account all observed follicles of different sizes is complex to interpret, and therefore the size of the largest or "lead" 2–3 follicles is often used as a surrogate (albeit less precise) measure to represent the entire cohort of follicles[12].

Definitive clinical consensus on the optimal size of these lead follicles is yet to be determined[13], such that IVF centers will generally use a threshold of either two or three lead follicles greater than 17 or 18 mm in diameter as a simple criterion to initiate trigger administration[10]. However, reliance on the lead follicle to represent the entire cohort is only appropriate if a single tight cohort of follicle sizes is recruited, and therefore lacks precision in comparison to consideration of the sizes of each follicle individually[10].

In the present study, we use XAI techniques examining all individual follicles to identify the size of follicles on the day of trigger administration that contribute most to the number of mature oocytes subsequently retrieved. These data can aid in identifying the optimal time to administer the trigger of oocyte maturation and, in turn, optimize downstream clinical outcomes. We also examine the impact of follicle size on premature progesterone elevation, which can advance maturation of the endometrium such that it is out of sync with the embryo at the time of transfer, to negatively impact live birth rates[14]. To this end, we leveraged data from 11 clinics across the United Kingdom and Poland incorporating the first treatment cycle from more than 19,000 patients. We have conducted the largest study to date to identify follicles that are most likely to yield mature oocytes, zygotes, and blastocysts, to optimize clinical outcomes and aid in the personalization of IVF.

## Results
### Follicle sizes that contribute most to clinical outcomes
Follicle sizes on the day of trigger (DoT) that contributed most to the number of mature oocytes subsequently retrieved on the day of OPU were identified using a histogram-based gradient boosting regression tree model[15]. Higher permutation importance values of the model input features indicated the most contributory follicle sizes. Follicles sized 12–20 mm on the DoT contributed relatively the most to the number of all oocytes retrieved (Fig. 1a) (n = 19,082), whereas follicles

sized 13–18 mm contributed most to the number of mature, i.e., metaphase-II (MII), oocytes retrieved (Fig. 1b) (n = 14,140).

When considering downstream laboratory outcomes such as two-pronuclear (2PN) zygotes (i.e., mature oocytes that have been fertilized), and high-quality blastocysts, similar follicle sizes remained most contributory (Fig. 1c, d). Specifically, follicles sized 13–18 mm contributed most to the number of 2PN zygotes (n = 17,822), (Fig. 1c), and follicles of 14–20 mm were the most important for high-quality blastocysts (n = 17,488).

As a sensitivity analysis, we selected intracytoplasmic sperm injection (ICSI) treatment cycles (n = 14,140), as only oocytes confirmed to be mature would proceed to an attempt at fertilization, and again follicles of size 13–18 mm were most contributory to yielding 2PN zygotes (n = 13,415), but a tighter range of 15–18 mm follicles were most likely to yield high-quality blastocysts (n = 12,091).

### Impact of age or treatment protocol on most contributory follicle sizes
We further investigated if the most contributory follicle sizes to the number of MII oocytes differed by age or the type of IVF treatment protocol used. Follicles sized 13–18 mm were again the most important in patients ≤35 years (n = 5707) (Fig. 2a), whereas in those >35 years (n = 4717), a broader category of follicles sized 11–20 mm was most contributory (Fig. 2b), of which follicles 15–18 mm provided the greatest contribution.

Next, in patients who received an hCG trigger, we evaluated whether the type of IVF treatment protocol used, i.e., GnRH agonist ("long", n = 5420) or GnRH antagonist ("short", n = 3981) co-treated protocols impacted on which follicle sizes were most contributory. Follicles sized 14–20 mm contributed most to MII oocytes in "long" protocol cycles (Fig. 2c), whereas follicles sized 12–19 mm were most important in "short" protocol cycles (Fig. 2d).

### Most contributory follicle sizes on days preceding trigger administration
To further validate findings and examine follicle sizes that were most contributory if the final ultrasound scan during OS was conducted prior to the DoT, we implemented two further models on data from patients who also had ultrasound scans on the penultimate day (DoT-1; n = 10,457 patients), or ante-penultimate day (DoT-2; n = 9533 patients), prior to the DoT (Fig. 3a). The most contributory size range was 12–16 mm on DoT-1 and 10–15 mm on DoT-2, which corresponds to expected mean follicle growth rates of 1–2 mm per day[16].

### Predictive model performances and explainability
Results from each trained model are presented in Table 1. For each model, the mean performance across all folds of cross-validation is reported with its standard deviation (SD). In general, after extensive hyperparameter tuning (Supplementary Table 2), the results across all models were similar. The units of mean absolute error (MAE) and median absolute error (MedAE) correspond to the absolute number of oocytes, MII oocytes, 2PN zygotes, or high-quality blastocysts. The model for predicting mature oocytes in the ICSI population (n = 14,140 patients) performed with MAE 3.60 (SD 0.35) and MedAE 2.59 (SD 0.31), averaged across the eleven clinics during the "internal-external validation" procedure. Performance of the models predicting all oocytes and MII oocytes are presented for each individual clinic in Supplementary Table 1, and a plot of predicted versus actual MII oocytes collected color-coded by each clinic is presented in Supplementary Fig. 1. Assessment of a multilayer perceptron model for predicting MII oocytes presented a higher MAE 3.85 (0.53), identifying 14–18 mm follicles as the most important. When excluding potential aberrant data for the ICSI population in a separate model predicting MII oocytes (n = 11,819), the average MAE improved to 2.54 (0.45) and $R^2$ to 0.49 (0.06).

Moreover, to verify the results further, a plot of "SHAP" values for one clinic is shown in Fig. 3b. An accentuated increase in "SHAP" values was observed across a similar range of intermediately-sized follicles, especially when three or more follicles of that size were present. This, in turn, corresponds to an increased expectation of MII oocytes. Conversely, when there were no counts of follicle sizes in the middle range, the "SHAP" values were accentuated negatively. Follicle sizes in the extrema correspond to minimal increases in the expectation of MII oocytes, even if there are many of them present, consistent with the analysis shown in Fig. 1b.

We evaluated the impact of different variables on the ability of the trained models to predict the number of MII oocytes retrieved. Whilst follicles of 13–18 mm had the highest relative contribution to the number of mature oocytes retrieved, this does not imply that other follicles were not also contributory, albeit to a relatively lesser extent. If we do not consider follicle size at all and include the total number of follicles on the DoT regardless of size (i.e., all follicles sized 6–26 mm), the resulting model had an MAE 3.92 (0.45) and $R^2$ 0.26 (0.15). When considering a single follicle size range, we found that the number of follicles sized 12–23 mm was most predictive (with respect to the lowest MAE) of the number of mature oocytes retrieved (MAE 3.71 (0.23); $R^2$ 0.31 (0.12)). However, our ML model, which considered the count of each individual follicle size, as well as the relative contribution of a follicle of that size, had even greater predictive performance (MAE 3.60 (0.35); $R^2$ 0.35 (0.13)) (Table 1). Incorporating other variables including: age, body mass index, days of stimulation, the type of trigger, IVF protocol, and estradiol on the DoT administration ($n = 2068$) as separate input variables, only improved the MAE by 0.06 units (MAE: 3.54 (0.36)) and $R^2$ by 0.01 ($R^2$: 0.36 (0.12)). Overall, knowledge of follicle size on the DoT was the most important factor in estimating the number of MII oocytes retrieved.

### Improvements in mature oocyte yield

We used data from all maturity-graded patients (ICSI cycles; $n = 14,140$ patients) to compare cohorts that fulfilled or did not fulfill the threshold-based criteria typically used as "rules-of-thumb" in current clinical practice (Fig. 4a). We calculated the percentage yield of MII oocytes divided by the number of all follicles (sized 6–26 mm) on the DoT (denoted here as the "mature oocyte yield"). The maximum positive median difference was noted as a 10% improvement (Mann-Whitney $U$-test: $p < 0.0001$) in mature oocyte yield when at least three follicles sized 17 mm were present on the DoT. The smallest improvement in mature oocyte yield was noted when two follicles of size 17 mm were present on the DoT ($p = 0.229$).

When comparing the follicle size threshold-based criteria to the proposed range-based criteria (Fig. 4b), having at least 10% of follicles in the range 13–18 mm (or 15–18 mm) on the DoT was associated with a >10% improvement in mature oocyte yield ($p < 0.0001$). Furthermore, maximizing the proportion of follicles on the DoT within the optimal size range could further improve mature oocyte yield, for example, by up to 42% when at least 70% of follicles were sized 15–18 mm ($p < 0.0001$).

### Impact of follicle size profile on live birth rate

Follicles sized 13–18 mm contributed most to the number of MII oocytes retrieved and therefore we analyzed whether the proportion of follicles within this size range was associated with the live birth rate (LBR) after fresh embryo transfer. LBR was 30.48% (95% CI: 29.68%–31.29%) in the patient cohort ($n = 12,724$). Using a logistic regression model ($n = 9209$), we found that the proportion of follicles within 13–18 mm on the DoT was positively associated with LBR (OR: 1.03 (1.00–1.06) per 10% points change; $p = 0.048$) when adjusted for age, total follicle count, and type of trigger administered (Fig. 5a). We next examined whether the mean follicle size impacted on

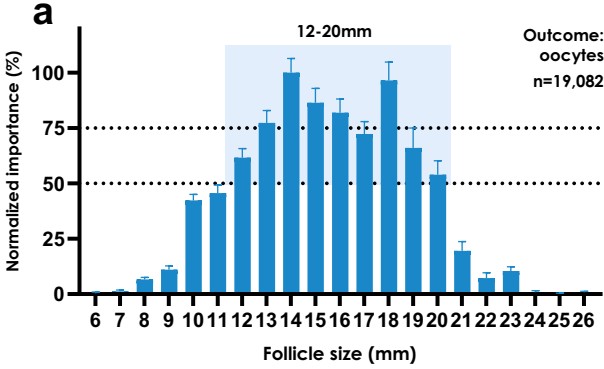

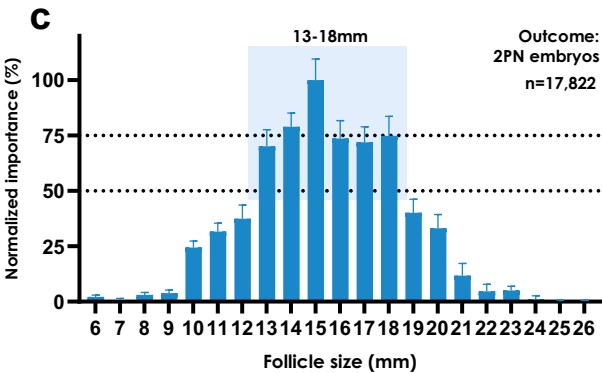

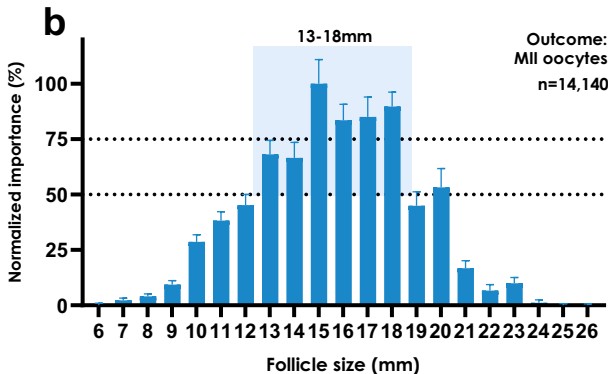

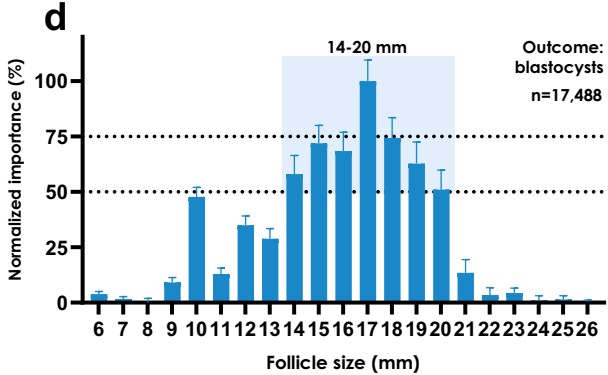

**Fig. 1 | Contributory follicle sizes for clinical outcomes.** Normalized permutation importance values (mean ± SD) of follicle sizes (in mm) in treatment cycles averaged across all eleven clinics in the cross-validation protocol. The outcome variables are all oocytes (**a**), metaphase-II (MII) mature oocytes (**b**), two-pronuclear (2PN) fertilized zygotes (**c**), and high-quality blastocysts (**d**), respectively. The shading highlights follicle sizes that have at least 50% normalized importance.

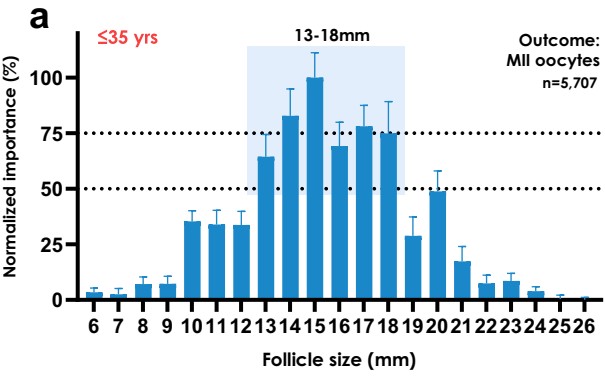

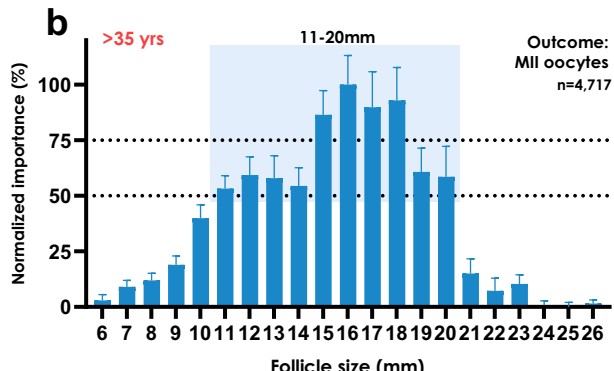

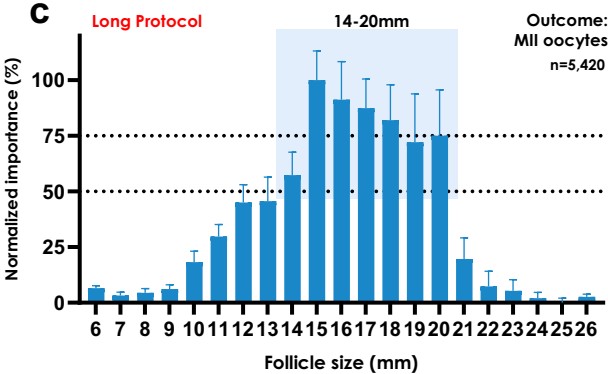

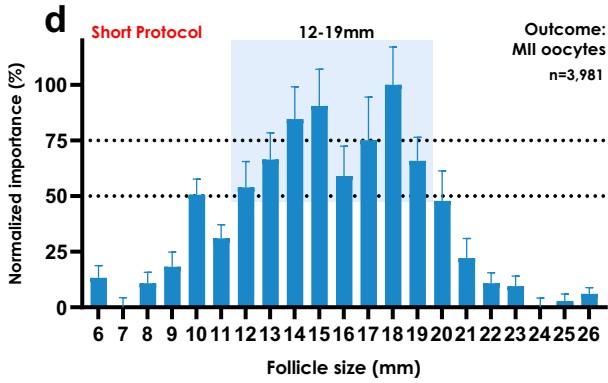

**Fig. 2 | Contributory follicle sizes for mature oocytes stratified by patient subgroups.** Normalized permutation importance values (mean ± SD) in treatment cycles of follicle sizes (in mm) in ICSI treatment cycles where oocyte maturity was graded (n = 14,140), averaged across all eleven clinics in the cross-validation protocol. The outcome variable is mature metaphase-II (MII) oocytes in all panels. The shading highlights follicle sizes that have at least 50% normalized importance. In a first stratification approach, (**a**, **b**) represent patients that were ≤35 years old at the time of treatment (n = 5707), and >35 years of age (n = 4717), respectively. The data in (**c**, **d**) are stratified by those with a GnRH agonist ("long"; n = 5420) or GnRH antagonist ("short"; n = 3981) IVF suppression protocol, respectively. In all cases, only cycles were considered where an hCG trigger was administered.

LBR (Fig. 5b), and found a negative association (OR: 0.95 (0.93–0.98) per 1 mm change; p = 0.001). We found that LBR (n = 427) was reduced with progesterone elevation on the DoT (Fig. 5c), whilst mature oocyte yield (n = 646) remained similar. Further, serum progesterone on the DoT (n = 994) was increasingly elevated as the number of follicles sized >18 mm on the DoT was greater (Fig. 5d).

## Discussion

We present findings from a large multi-center European study incorporating data from 19,082 patients utilizing XAI techniques to establish the relationship between follicle sizes at the end of ovarian stimulation (OS) and the subsequent retrieval of mature oocytes, as well as relevant downstream clinical outcomes. We showed that a contiguous range of follicles sized 13–18 mm contributed most to the number of mature oocytes subsequently retrieved. In comparison to using lead follicle size to inform trigger timing, our data suggests that maximizing the proportion of follicles within this size range could further optimize the number of mature oocytes retrieved to improve IVF outcomes, including LBR, pending prospective evaluation. Furthermore, extending the duration of OS resulted in a greater number of larger follicles (>18 mm) on the DoT that not only contributed less to the yield of mature oocytes but also resulted in premature progesterone elevation with a consequent negative impact on live birth[17,18], likely due to its adverse effect on the endometrial stage[14]. These data highlight the potential of ML methods to aid in personalizing ART treatment to optimize clinical outcomes.

Our results are consistent with a previous pilot study using a much smaller sample size of 499 patients that found that follicles sized 12–19 mm on DoT were most contributory to the number of mature oocytes[10]. Hariton et al. found that an input feature containing the number of follicles sized 16–20 mm on the DoT was most contributory to the model performance using an ensemble ML model of similar complexity in data from a single clinic comprising 7866 ICSI treatment cycles with various protocol types[19]. Similarly, Reuvenny et al. used an XGBoost model on data from GnRH antagonist ("short" protocol) co-treated cycles from a single center (n = 3599 treatment cycles) to show that an input variable of 14–16 mm sized follicles on the DoT highly contributed towards model performance[20]. Further, Fanton et al. developed a linear regression model with data from three clinics in the USA (n = 30,278 treatment cycles) that identified an input variable of 14–15 mm sized follicles on the DoT as the most important, followed by 16-17 mm follicle sizes[21]. In our study, we examined individual follicle sizes to identify the group of follicles that contribute most to the retrieval of mature oocytes and, in turn, downstream clinical outcomes.

To validate the methodology further, we also analyzed follicle sizes that were most important on the days prior to DoT. We showed that follicles sized 10–15 mm on DoT-2 and 12–17 mm on DoT-1 were the most contributory to predicting MII oocyte yield on the days preceding DoT. This smooth trajectory of optimal follicle size range shift corroborated well with expectations in mean follicular growth rates per day during OS[16]. The aforementioned ML studies also showed agreement, where the most important input variables to the models utilizing follicle sizes dropped by 1–2 mm in range a day prior to the DoT[20,21].

In order to examine whether similar follicle sizes were important in predicted good ovarian response patients, we stratified our data by age. We found that follicles sized 13–18 mm were most contributory in

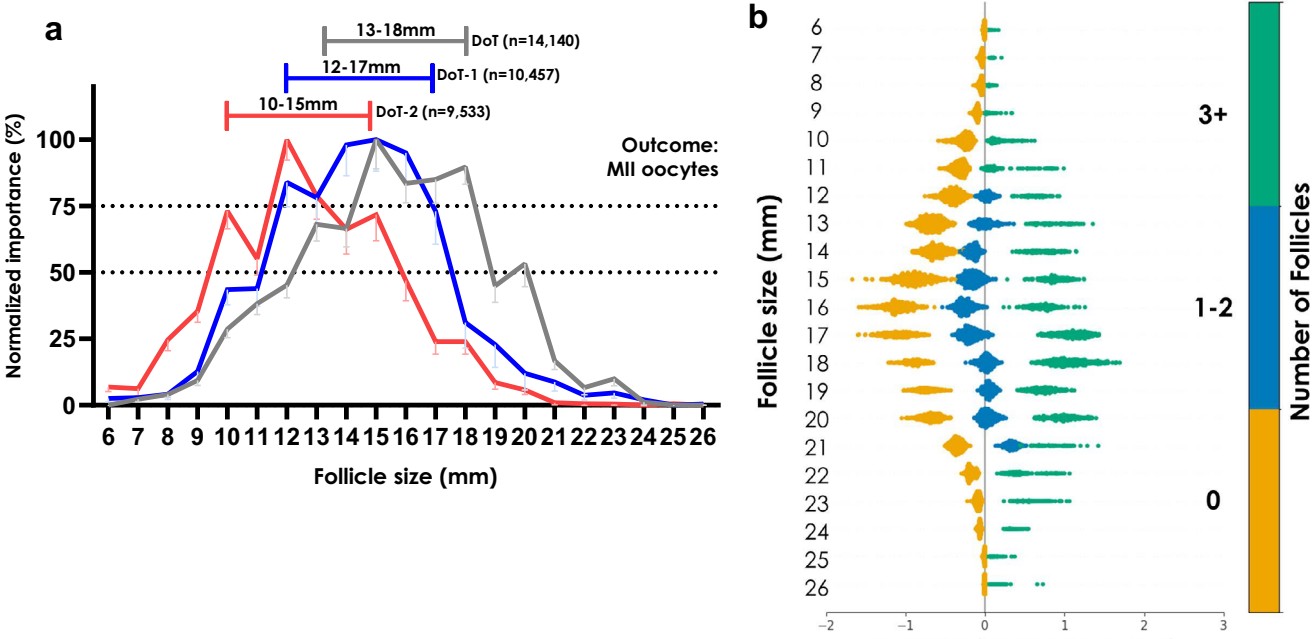

**Fig. 3 | Contributory follicles on preceding days to trigger and explainability on the day of trigger. a** Normalized permutation importance values (mean ± SD) in treatment cycles of follicle sizes (in mm) in ICSI treatment cycles where oocyte maturity has been graded averaged across all eleven clinics in the cross-validation protocol. Three separate models were trained on treatment cycles where an ultrasound scan was available on the day of trigger (DoT) administration (n = 14,140), a day prior to trigger (DoT-1; n = 10,457), or two days prior to trigger (DoT-2; n = 9533). The figure represents the expected growth trajectory of the most important follicle sizes during ovarian stimulation approaching the DoT. **b** A beeswarm plot of "SHAP" values for one clinic in the cross-validation protocol, indicating the contribution to the predicted value of metaphase-II (MII) oocytes. The discrete color map corresponds to the count of follicles on the DoT administration in each treatment cycle in the clinic; yellow: no (zero) follicles, blue: 1–2 follicles, green: 3 or more follicles, of that particular size (in mm).

**Table 1 | Model performances for predicting clinical outcomes**

| Outcome variable | All oocytes | MII oocytes | 2PN zygotes | | HQ blastocysts | |
| --- | --- | --- | --- | --- | --- | --- |
| Patient group | All | ICSI | All | ICSI | All | ICSI |
| Number of patients/cycles | 19,082 | 14,140 | 17,822 | 13,415 | 17,488 | 12,091 |
| MAE | 3.85 (0.27) | 3.60 (0.35) | 2.90 (0.36) | 2.89 (0.36) | 1.94 (0.26) | 1.89 (0.27) |
| MedAE | 2.68 (0.15) | 2.59 (0.31) | 2.07 (0.22) | 2.08 (0.21) | 1.38 (0.15) | 1.39 (0.18) |
| RMSE | 5.56 (0.61) | 5.06 (0.60) | 4.12 (0.60) | 4.10 (0.63) | 2.79 (0.47) | 2.71 (0.48) |
| $R^2$ | 0.45 (0.13) | 0.35 (0.13) | 0.25 (0.21) | 0.24 (0.21) | 0.08 (0.11) | 0.08 (0.12) |
| Max error | 38.07 (10.16) | 31.84 (10.39) | 26.84 (9.46) | 25.68 (10.05) | 17.82 (3.93) | 16.35 (4.85) |
| Outcome variable | MII oocytes (ICSI patients) | | | | | |
| Patient group | ≤35 yrs | >35 yrs | Long protocol | Short protocol | DoT-1 | DoT-2 |
| Number of patients/cycles | 5707 | 4717 | 5420 | 3981 | 10,457 | 9533 |
| MAE | 4.09 (0.47) | 3.58 (0.65) | 3.23 (0.26) | 4.17 (0.92) | 3.51 (0.39) | 3.63 (0.37) |
| MedAE | 3.01 (0.38) | 2.60 (0.53) | 2.45 (0.35) | 3.13 (0.72) | 2.47 (0.27) | 2.59 (0.38) |
| RMSE | 5.66 (0.86) | 5.00 (0.97) | 4.41 (0.44) | 5.72 (1.22) | 4.94 (0.59) | 5.15 (0.62) |
| $R^2$ | 0.28 (0.14) | 0.33 (0.15) | 0.35 (0.09) | 0.35 (0.18) | 0.36 (0.13) | 0.34 (0.14) |
| Max error | 29.74 (8.08) | 26.68 (9.60) | 21.39 (7.16) | 24.79 (5.65) | 30.16 (8.56) | 29.86 (10.88) |

Model performances, mean (SD), after nested leave-one-clinic-out cross-validation. The outcome variable corresponds to the target to be predicted (e.g., mature oocytes), using follicle size ultrasound data from the day of trigger (DoT). Data were sourced from eleven clinics and all numbers of patients also represent the number of treatment cycles. All performance metrics were calculated and reported on the outer folds of the internal-external cross-validation procedure.

*ICSI* intracytoplasmic sperm injection, *MAE* mean absolute error, *MedAE* median absolute error, *RMSE* root-mean-square error, $R^2$ coefficient of determination, *MII* metaphase-II, *2PN* two pronuclear, *HQ* high-quality.

younger patients aged ≤35 years, but those 11–20 mm (particularly 15–18 mm) were most important in patients >35 years. In line with our data whereby increased mean follicle size by the end of OS was associated with a negative impact on LBR, it has been suggested that older patients with diminished ovarian reserve may benefit from earlier trigger administration in modified natural cycles[22]. To date, the impact

of age on trigger timing in IVF cycles remains uncertain. Likewise, although most patients are required to be non-smokers to access state-funded care, there was insufficient recording of smoking status to formally assess any impact of smoking.

It has been hypothesized that due to the preceding period of gonadotropin suppression that could synchronize follicle growth, the

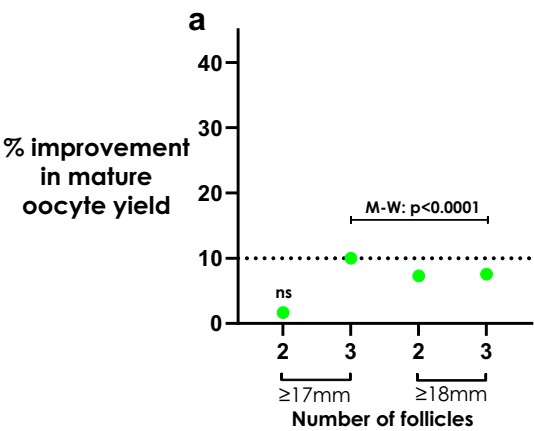

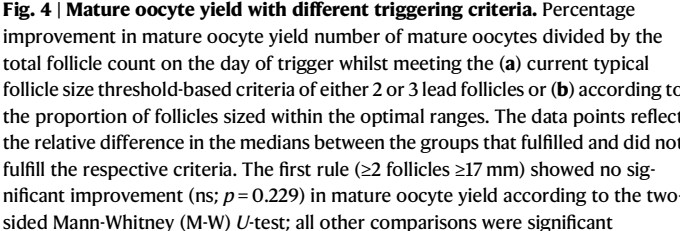

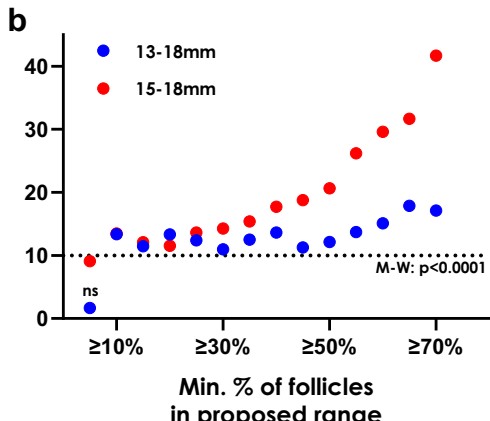

**Fig. 4 | Mature oocyte yield with different triggering criteria.** Percentage improvement in mature oocyte yield number of mature oocytes divided by the total follicle count on the day of trigger whilst meeting the (**a**) current typical follicle size threshold-based criteria of either 2 or 3 lead follicles or (**b**) according to the proportion of follicles sized within the optimal ranges. The data points reflect the relative difference in the medians between the groups that fulfilled and did not fulfill the respective criteria. The first rule (≥2 follicles ≥17 mm) showed no significant improvement (ns; $p = 0.229$) in mature oocyte yield according to the two-sided Mann-Whitney (M-W) $U$-test; all other comparisons were significant

($p < 0.0001$). Aside from the first threshold-based rule, all other rules in (**a**) showed significant yield improvement compared to patients not meeting the respective criterion by the M-W $U$-test ($p < 0.0001$). In (**b**), all range cut-offs ≥5% showed statistically significant improvements in mature oocyte yield when using 15–18 mm as the criterion ($p < 0.0001$). All range cut-offs ≥10% showed significant improvements in mature oocyte yield when using 13–18 mm as the criterion ($p < 0.0001$); ≥5% of follicles in the 13–18 mm range presented no significant improvement in mature oocyte yield (ns; $p = 0.125$).

GnRH agonist co-treated ("short") protocol could lead to a more uniform cohort of follicles than in the GnRH antagonist co-treated ("long") protocol[23]. We found that larger follicles sized 14–20 mm were more contributory to the yield of mature oocytes in GnRH agonist co-treated protocol cycles whereas follicles sized 12–19 mm were most important in GnRH antagonist protocol cycles. A meta-analysis of seven randomized control trials ($n = 1295$ patients) comparing protocol types found that patients undergoing the "long" protocol had a significantly higher number of oocytes retrieved ($p < 0.00001$) in those that received a delayed hCG trigger by 24–48 h after OS[24]. Our data are consistent with these previous trials, suggesting that larger follicles may be associated with improved oocyte yield in the GnRH agonist co-treated protocol[25,26].

Several points of strength should be highlighted in our presented study. Firstly, we used XAI techniques in our study. Explainability is currently a paramount characteristic of AI decision-making in ART clinics and the introduction of data-driven insights that align with clinical reasoning promotes the possible wider adoption of clinical decision-support systems in the ART domain[2,27,28]. Secondly, the use of data across two countries comprising eleven clinics presents a heterogeneous patient population with a variety of clinical practices and treatment protocols. Since many clinics are involved, the use of internal-external validation at this stage of development was appropriate[29,30], whereby the developed models and their performance metrics are provided with a standard deviation due to each clinic behaving as an independent test set to observe permutation importance and model error. Furthermore, we chose to avoid random splits of data from the same clinics in both training and test sets when reporting outcome metrics, as this has been shown to result in bias and unknown generalizability ("transportability") to wider patient populations[30]. Thirdly, by only using individual follicle sizes as separate input variables (i.e., not in a grouped/binned fashion), a contiguous range of follicles that contribute most towards mature oocyte yield was more explicitly identified. Finally, we ensured only to include the first treatment cycle of each patient in this study for model development and validation (i.e., $n = 19,082$ represents both the number of treatment-naive patients and cycles). This was to avoid auto-correlation between successive cycles of a single patient (e.g., a clinician's decision-making is likely to be influenced by a previous

treatment cycle) since longitudinal random effects across sequential cycles can influence the permutation importance[31]. Similarly, input parameters that present multi-collinearity (e.g., estradiol and follicle size[32]) can also result in unreliable insights from permutation importance analysis.

Variability in follicle size measurements, both within and between observers, has been a documented challenge in ART[33]. We observed a few cases where the total number of oocytes retrieved exceeded the follicles recorded at the time of trigger. This discrepancy underscores the likelihood that smaller follicles are often not consistently recorded, a practice echoed in anecdotal discussions with clinicians, who frequently attribute this to the lower probability of such follicles yielding oocytes. A previous study has shown that follicle imputation has limited impact on enhancing model performance in this specific context[21]. Although excluding such treatment cycles could artificially improve model error, it is then not possible to build models that are more robust to such measurement error in future applications. Our approach utilizes an ensemble-based ML model with Bayesian optimization and can mitigate against the impact of data inconsistencies due to extremes of biological variation and/or measurement error[33]. By employing a loss metric like mean absolute error (MAE), which is less sensitive to outliers, the model offers a more robust analysis validated across multiple clinics. The need for more objective ultrasound scanning methods, potentially through the integration of automated algorithms, may further improve the accuracy and reliability of follicle measurements and associated algorithms in ART[33].

Our data suggest that a novel approach to deciding when to administer the trigger of oocyte maturation could be based on the proportion of intermediately-sized follicles (e.g., 13–18 mm) rather than the traditional threshold-based approach assessing when 2–3 lead follicles reach 17 or 18 mm in size (Fig. 4). Further, our data also suggests that mean follicle size could impact LBR in fresh embryo transfer cycles (Fig. 5), aside from a direct effect on the ability to yield oocytes, potentially via larger follicles resulting in premature progesterone elevation[34,35]. A prospective randomized controlled trial is required to demonstrate the benefit of our new proposed approach to determine trigger administration based on the entire follicle cohort in comparison to the current threshold-based approach.

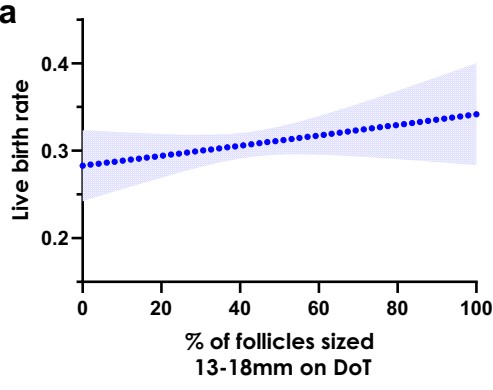

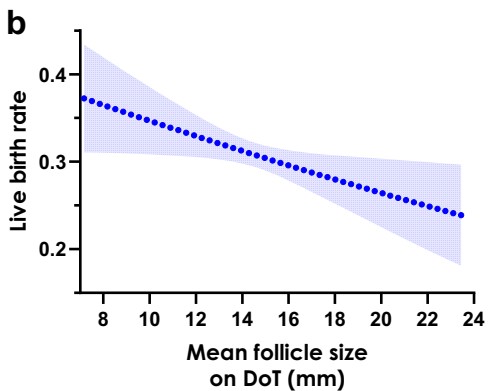

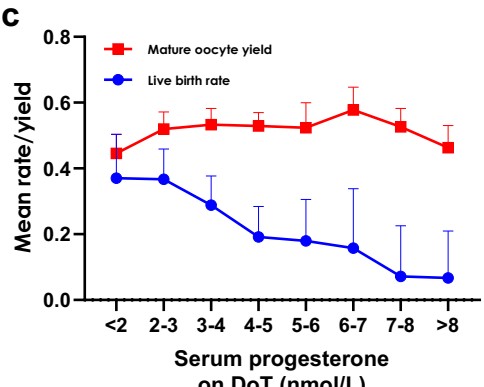

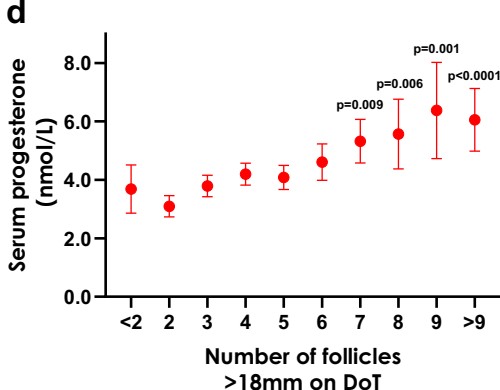

**Fig. 5 | Impact of follicle size profile and elevated progesterone on live birth rate.** Partial dependence plots under logistic regression modeling (*n* = 9843) to demonstrate the impact of (**a**) the percentage of follicles sized 13–18 mm on the day of trigger (DoT) (OR: 1.03 (1.00 − 1.06) per 10 percentage points change; *p* = 0.048), and (**b**) mean follicle size on the DoT (OR: 0.95 (0.93 − 0.98) per 1 mm change; *p* = 0.001), on predicted live birth rate (LBR). Both models were adjusted for: age, total follicle count on the DoT, and type of trigger administered (hCG or GnRH agonist). The mean LBR and its 95% confidence interval (CI) are plotted with 100 bootstrapped simulations. **c** The association (mean and 95% CI) between increasing progesterone elevation on the DoT versus mature oocyte yield (*n* = 646) and LBR (*n* = 427). **d** The association (mean and 95% CI) between an increasing number of follicles sized >18 mm on DoT versus serum progesterone levels on the DoT (*n* = 994). Having at least 7 follicles of >18 mm in size presented a significant elevation in progesterone with reference to patients with <2 follicles >18 mm on the DoT using the two-way Dunnett's multiple comparisons test reported with adjusted *p* values.

Since our data demonstrates a possible trade-off between LBR and delayed trigger administration, a trial comparing trigger strategies on this basis would therefore be potentially more definitive in determining the impact of trigger timing on clinical outcomes. It should be noted that although a range of follicle sizes that contribute relatively more than others varied marginally depending on the patient stratifications considered (Fig. 2). Ultimately, an ML model that considers individual follicle sizes and their relative contributions, in addition to patient characteristics, could be harnessed as part of a clinical decision support system[2,28].

In conclusion, we establish that intermediately-sized follicles on the day of trigger contribute the most to the retrieval of mature oocytes and subsequent embryo development. Utilizing the sizes of all follicles, rather than just the size of only the lead follicles, could offer a target for OS protocols and inform the timing of trigger administration to optimize clinical outcomes. These data highlight the potential of XAI techniques to provide data-driven optimization of IVF treatment to improve clinical outcomes.

## Methods
### Study participants
We used data from a multi-ethnic cohort of 19,082 female participants who had treatment in one of eleven clinics geographically distributed across the United Kingdom (nine) and Poland (two) between 2005 and 2023 (Table 2). We selected treatment-naive patients who had a

transvaginal ultrasound scan presenting at least three follicles >10 mm on the same day as the DoT administration. This cohort was 18–49 years of age at the time of treatment with a median body mass index (BMI) of 24.17 kg/m² and antral follicle count of 15.00. Where available, an assessment of oocyte maturity grade detailing metaphase-II oocytes (*n* = 14,140 patients) was used as the primary outcome (i.e., in ICSI treatment cycles). Downstream outcomes such as the number of 2PN zygotes (*n* = 17,822 patient cycles), and high-quality blastocysts (*n* = 17,488 patient cycles), were also assessed. Where data were unavailable in patients electronic health records for specific demographic information or clinical outcomes, these were excluded from the respective analyses and noted in Table 2.

### Study approvals
Data included in this manuscript were obtained from a retrospective study carried out following the sponsorship of the institutional review board and approval by the Health Research Authority (23/HRA/2849). All subjects gave written informed consent in accordance with the Declaration of Helsinki and Good Clinical Practice. All ART clinics were under a license from the Human Fertilization and Embryology Authority (UK) or the Ministry of Health (Poland).

### In vitro fertilization protocol
This was a retrospective cohort study analyzing follicle and oocyte data from IVF or ICSI cycles. The objective was to identify the follicle

**Table 2 | Patient demographics and treatment information**

| Variable | Min | LQ | Median | Mean | UQ | Max | IQR | SD | N | Missing |
|---|---|---|---|---|---|---|---|---|---|---|
| Age at treatment (yrs) | 18.20 | 31.60 | 34.70 | 34.61 | 38.00 | 49.00 | 6.40 | 4.50 | 19,080 | <1% |
| Body mass index (kg/m$^2$) | 19.00 | 21.72 | 24.17 | 24.98 | 27.51 | 47.25 | 5.79 | 4.25 | 10,965 | 43% |
| Antral follicle count | 1.00 | 8.00 | 15.00 | 17.39 | 23.00 | 130.00 | 15.00 | 13.28 | 9620 | 50% |
| Days to trigger | 4.00 | 10.00 | 11.00 | 11.34 | 12.00 | 32.00 | 2.00 | 1.99 | 19,072 | <1% |
| No. follicles on DoT | 3.00 | 9.00 | 14.00 | 16.16 | 20.00 | 90.00 | 11.00 | 9.86 | 19,082 | 0% |
| No. oocytes collected | 0.00 | 7.00 | 11.00 | 12.54 | 16.00 | 86.00 | 9.00 | 7.78 | 19,082 | 0% |
| No. MII oocytes | 0.00 | 5.00 | 8.00 | 9.62 | 13.00 | 68.00 | 8.00 | 6.59 | 14,140 | 26% |
| No. 2PN zygotes | 0.00 | 3.00 | 6.00 | 7.15 | 10.00 | 58.00 | 7.00 | 5.18 | 17,822 | 7% |
| No. HQ blastocysts | 0.00 | 1.00 | 2.00 | 3.17 | 4.00 | 31.00 | 3.00 | 3.05 | 17,488 | 8% |
| Mature oocyte yield | 0.00 | 0.40 | 0.59 | 0.66 | 0.83 | 5.86 | 0.43 | 0.41 | 14,140 | 26% |
| Maturation rate | 0.00 | 0.67 | 0.80 | 0.76 | 0.92 | 1.00 | 0.26 | 0.23 | 14,140 | 26% |
| Fertilization rate | 0.00 | 0.55 | 0.71 | 0.69 | 0.86 | 1.00 | 0.31 | 0.22 | 12,757 | 33% |
| Blastulation rate | 0.00 | 0.29 | 0.50 | 0.48 | 0.67 | 1.00 | 0.38 | 0.29 | 16,484 | 14% |
| Live birth rate | - | - | - | 30.48% | - | - | - | - | 12,724 | 33% |

Patient demographics and treatment cycle information from the total population of 19,082 participants. Data are reported as the minimum, lower quartile (LQ), median, mean, upper quartile (UQ), maximum, standard deviation (SD), and interquartile range (IQR). Mature oocyte yield is defined as the number of metaphase-II (MII) mature oocytes collected divided by the total follicle count on the day of trigger (DoT) administration.
*No.* number of, *2PN* two-pronuclear, *HQ* high-quality.

sizes on the DoT that are most likely to yield mature oocytes and therefore provide a target for ovarian stimulation and expected oocyte number.

Patients at elevated risk of ovarian hyperstimulation syndrome (OHSS) or who are noted to have premature progesterone elevation are often advised to have their embryos cryopreserved pending a frozen embryo transfer (called "freeze-all")[3]. A freeze-all strategy can mitigate the risk of reduced implantation due to premature progesterone elevation, albeit at the expense of increasing the time-to-pregnancy and the risk of perinatal complications, e.g., large-for-gestational-age babies[14]. For analysis of live birth in this study, women underwent either IVF or ICSI with fresh embryo transfer and had their final ultrasound scan to assess follicle size on the DoT administration. Follicle growth was induced using daily preparations containing FSH activity. Patients underwent a suppressant protocol to prevent premature ovulation using either a gonadotropin-releasing hormone (GnRH) agonist ("long" protocol; $n = 6990$) or antagonist ("short" protocol; $n = 7408$) co-treated protocol. Following this, a trigger of either human chorionic gonadotropin (hCG; $n = 13,473$) or GnRH agonist ($n = 1675$) was administered generally once two to three follicles had reached 17 or 18 mm in diameter (the clinician's decision).

Only the first recorded IVF cycle of the patient was used in all analyses (i.e., treatment-naive patients). Primary analysis was carried out on patients who had an ultrasound scan on the DoT. Laboratory results, including assessments of oocyte maturity, embryo quality, and blastocyst quality, were included as outcome measures. Subsequent analyses were carried out on patients with ICSI treatment only and ultrasound scans available on the penultimate (DoT-1; $n = 10,457$) or ante-penultimate (DoT-2; $n = 9533$) days before the DoT as further methodological validation.

### Statistical analysis

**Data pre-processing.** The ultrasound scans during OS of IVF/ICSI treatment were used to obtain the follicle sizes for each patient. 2D-scanning of the follicles provides a diameter measurement in millimeters, which were recorded to integer precision by ultrasonographers in electronic health records. Follicle diameters of the same size were therefore grouped and counted in 1 mm increments from 6 to 26 mm on the DoT. We used the individual follicle sizes as input variables:

$$\mathbf{X} = X_6, \ldots, X_{26}$$
$$\text{where} \quad X_6 = \text{number of follicles sized 6 mm,}$$
$$X_7 = \text{number of follicles sized 7 mm,}$$
$$\text{and so on.} \tag{1}$$

For the outcome measure, the number of all oocytes ($y_{ooc}$), and more specifically MII oocytes retrieved (representing a subset of all oocytes capable of fertilization), were used in the regression models. To normalize the right skewness, the outcome was transformed using the natural logarithm:

$$y_{out} = \ln(y_{ooc} + 1) \tag{2}$$

The inverse transformation was carried out (3) during model testing and evaluation. In additional analyses, the number of 2PN zygotes and the number of high-quality blastocysts were used as the outcome measure with the same transformation strategy. Since patients often undergo several ART cycles, we ensured that only the first recorded cycle per patient was included in the dataset, so as to guard against intra-individual correlation introduced by the availability of a longitudinal information pool to inform decision-making in successive treatment attempts[31].

$$y_{ooc} = \exp(y_{out}) - 1 \tag{3}$$

**Model development and validation.** Several histogram-based gradient boosting regression tree models were trained[15]. This is an open-source library inspired by LightGBM (Microsoft), with much faster building procedures using histogram data structures. We employed leave-one-clinic-out cross-validation (LOCO-CV) procedure to train, validate, and test the models (so-called "internal-external validation")[29]. The mean absolute error (MAE) was optimized as an objective function using nested LOCO-CV with Bayesian optimization to tune relevant hyperparameters in the search space demonstrated in Supplementary Table 2. This procedure is where at every LOCO-CV fold, the eleventh (clinic) fold represents an independent test set. Within the other ten folds, the tenth is a validation set for tuning hyperparameters, and the remaining nine are used as a training set

under ten-fold cross-validation. The MAE was chosen as it is less sensitive to outliers and intuitively demonstrates the error in the model since the absolute error can be interpreted as a unit oocyte of loss.

Ten independent model pipelines were implemented with various output measures including oocytes, MII oocytes, 2PN zygotes, and high-quality blastocysts. Further model stratifications included age and IVF protocol type ("long" GnRH agonist or "short" antagonist). To ensure that the conclusive follicle size range was not impacted by using a subset of the patient cohort to analyze MII oocytes collected (where maturity grading was available), we trained models to compare the results for all oocytes collected in both cohorts (i.e., 19,092 patients of which 14,140 had ICSI treatment). Furthermore, to investigate the impact of outliers and potential aberrant data, we restricted the dataset in a separate model to cycles with 1–30 MII oocytes retrieved, and where the number of follicles on the DoT was at least equal to the number of MII oocytes retrieved ($n = 11,819$ patients).

Utilizing the same LOCO-CV procedure, we incorporated further input variables of interest such as age, BMI, days of stimulation, type of IVF protocol ("long" GnRH agonist or "short" GnRH antagonist), estradiol on the DoT, and the type of trigger administered (hCG or GnRH agonist), to observe whether the predictive capability of the mature oocytes model improved. We compared whether the MAE and $R^2$ notably improved to solely using the number of follicle sizes on the DoT as input.

Similarly, to observe any notable impact in the trade-off between model complexity and explainability, we modeled the primary outcome of MII oocytes ($n = 14,140$ patients) using a multilayer perceptron model (a shallow artificial neural network) and reported its MAE.

**Identifying the most contributory follicle sizes.** Explainability is a current priority in ART, and clinicians generally prefer to avoid black-box treatment recommendations[2,36]. Ensemble methods, therefore, offer a valuable trade-off in handling non-linear and complex underlying data, accompanied by explainable insights. Only once each model was trained and validated (using up to ten folds in total), was then the mean and standard deviation determined across five runs of the permutation importance of features using the eleventh independent test set in the LOCO-CV protocol[37]. In this paradigm, features are randomly shuffled to see their impact on model loss (here set as the MAE). To identify the key follicles that yield mature oocytes, we used a threshold of ≥50% normalized contribution to the model to indicate relative importance; as described in previous literature, we hypothesized this follicle size range of utmost utility to be contiguous[5,10]. To establish further insights from the data, we also analyzed patients who had a final ultrasound scan on the penultimate ($n = 10,457$) and ante-penultimate ($n = 9,533$) days prior to the DoT administration. We used these patient cohorts to identify if a step-wise trajectory in the size of important follicles was observed.

**Predictive capabilities and model explainability.** The Shapley Additive exPlanations ("SHAP") package, an alternative explainability method grounded in game theory[38], was used to provide a further interpretation perspective and reinforce our findings from the permutation importance analysis. As opposed to observing changes in model loss, the SHAP paradigm considers the coalition of features to estimate the contribution of each feature towards the predicted value, which can be positive or negative, in units of the loss function (i.e., unit MII oocytes). The "TreeSHAP" package is optimized for tree-based models and approximates the marginal expectation of the outcome and the contribution of each feature[39].

Separately, to identify which single follicle range was most predictive of mature oocytes, we evaluated the predictive ability of specific size ranges using univariable linear regression with LOCO-CV across the eleven clinics ($n = 14,140$). Then, to identify which range was

most predictive of the number of mature oocytes retrieved, we compared all possible follicle size ranges using the same method, optimized for MAE as the loss function.

**Determining improvements in mature oocyte yield.** We considered the cohort of patients who had ICSI ($n = 14,140$) where the number of MII oocytes was recorded to compare the threshold-based criteria currently used in clinical practice, and a proposed approach based on maximizing the proportion of follicles within the optimal size range. For each of the four variations of the typically used threshold-based criteria to determine the DoT administration (i.e., two or three follicles greater than size 17 or 18 mm), we assessed whether each patient cycle had fulfilled this criterion or not, and grouped them accordingly. We compared the relative difference in medians of mature oocyte yield (number of mature oocytes divided by the total follicle count on the DoT) in these two groups of patients and compared the subgroups using the Mann-Whitney $U$-test (Fig. 4a).

For the range-based criteria, we ran the same analysis at different minimum cut-offs of the percentage of follicles within that follicle size range on the DoT (e.g., ≥5%, ≥10%, and so on), to examine any improvement in mature oocyte yield when maximizing the follicle sizes in this range (Fig. 4b). All statistical comparisons were carried out using the two-sided Mann-Whitney $U$-test.

**Associations with live birth rates.** To determine any associations between the proposed follicle size and live birth rate (LBR), we used logistic regression on all data with LBR recorded and input variables ($n = 9843$) including the percentage of follicles in the proposed follicle size range, total follicle count on the DoT, age at the time of treatment, and type of trigger administered (hCG or GnRH agonist). We then repeated this by replacing the follicle size range input variable with the mean follicle size on the DoT follicle profile. We utilized 100 boot-strapped simulations to determine a 95% confidence interval (CI) for partial dependence (marginal contribution) of each variable, highlighting any statistical significance and their associations with LBR.

Further, we plotted the mean and 95% CI of the mature oocyte yield ($n = 646$) and LBR ($n = 427$) according to serum progesterone levels (nmol/L) on the DoT in 1 nmol/L increments (Fig. 5c). We compared serum progesterone on the DoT ($n = 994$) according to the number of follicles sized larger than the proposed optimal follicle size range (Fig. 5d). Progesterone levels were compared to those with less than two larger follicles using the Dunnett's multiple comparison test with adjusted $p$ values.

### Reporting summary
Further information on research design is available in the Nature Portfolio Reporting Summary linked to this article.

## Data availability
The data that support the findings of this study were obtained from TFP Fertility under a legal non-disclosure agreement and non-commercial terms. Due to the nature of these agreements, the data are not publicly available and cannot be shared by the researchers with third parties. Any requests for access to the raw data should be directed to the Business Intelligence team at TFP Fertility (bi-support@tfp-fertility.com). Requests should specify the principal investigator, a description of the proposed study, and the reason for data access. The proposed ML models and statistical pipelines openly available in the code are not specific to the datasets used in this study. A source data file is provided with this paper. Source data are provided with this paper.

## Code availability
The codes used to train the models and further statistical analyses are available in this repository: https://github.com/hanassabio/XAI_

follicle_sizes. If the code is used or repurposed, please cite the manuscript.

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

## Acknowledgements

The Department of Metabolism, Digestion, and Reproduction is funded by grants from the MRC and NIHR Imperial Biomedical Research Centre. S.H. is supported by the UKRI CDT in AI for Healthcare http://ai4health.io

(EP/SO23283/1). A.Ab. is supported by an NIHR Clinician Scientist Award (CS-2018-18-ST2-002). W.S.D. is supported by an NIHR Senior Investigator Award (NIHR202371).

## Author contributions

A.Ab., S.H., W.S.D., and T.H. conceived the study. S.H., A.Ak., and T.W.K. conceived the methods and designed the analytical approach. R.S. and G.H.T. enabled clinical data acquisition. S.H. and A.Ak. performed the computational analysis with clinical direction from A.Ab., S.M.N., and A.C.Y. S.H., A.Ab., and S.M.N. wrote the paper, and all authors read, reviewed, and approved the final manuscript.

## Competing interests

S.H. provides consultancy services for Impli Limited. S.M.N. received grants from NIHR, CSO, and Wellbeing of Women; provided consultancy services for Access Fertility, Ferring Pharmaceuticals, Roche, Ro, and TFP; received honoraria from Ferring Pharmaceuticals, Merck, and Roche; received support for attending meetings and/or travel from Ferring Pharmaceuticals, Merck, and Gideon Richter; and leadership role in the HFEA. W.S.D. received grants from NIHR, MRC, and Imperial Health Charity, and is a consultant for Myovant Sciences Ltd. A.A. has received grants from the BRC; has provided consulting services for Myovant Sciences Ltd; and received support for travel from Merck. The remaining authors declare no competing interests.
