## [Transparent Peer Review file · Nature Communications]

Explainable artificial intelligence to identify follicles that optimize clinical outcomes during assisted conception

Corresponding Author: Dr Ali Abbara

Version 0:

Reviewer comments:

Reviewer #1

(Remarks to the Author)

The manuscript by Hanassab et al. used explainable artificial intelligence (XAI) to identify follicle sizes that contribute most to IVF cycle outcomes. For this end, the authors performed a multicenter study including 11 European countries with 19,082 patients. They concluded that follicles in the size range of 13-18 mm on the day of trigger contribute the most to the retrieval of mature oocytes and subsequent embryo development and that maximizing this proportion of follicles by the end of ovarian stimulation was associated with improved live birth rates. Furthermore, they suggested that larger mean follicle sizes, especially those >18 mm, were associated with progesterone elevation by the end of the ovarian stimulation with a possible negative impact on live birth rates.

The topic, which has been previously published, is interesting and may aid physicians in personalization and optimization of cycle outcomes. Its main advantages are the large sample size, its generality representing different clinics and practices and its explainability which may contribute to its acceptance in daily use. Although there are already several publications on trigger day algorithms, the study is novel in its approach. However, several comments and concerns need to be addressed by the authors.

Comments:

1. The authors trained 11 different models by repeatedly selecting 1 clinic to act as test set, and using the other 10 clinics to train the model. It is unclear if during hyper-parameter tuning the pipeline had access only to the 10 clinics data, or it was able to repeatedly test itself against the final left out clinic's data. A more elaborate explanation on the training, validation and testing methodology should be supplied.
2. My major concern is the average performance of the article models in predicting M2 oocytes on day of trigger (MAE 3.6, R2 0.35) that underperforms compared to similar models in existing publications (Reuveny - MAE 2.2, R2 0.69 ; Fanton – 2.87, R2 – 0.64), this weakens the results of the article, and should be discussed.
3. The article focus on the effect of follicle size on the clinical outcome. It is known that many other parameters such as estradiol levels might be involved. It is not clear what was their impact on the model, which might weaken the effect of a model relying on the follicular size only. This effect should be reported and discussed.
4. The article states that the high importance of some features (follicles 13-18) in their specific models is an indication that they are important in general for optimizing the number of oocytes retrieved, this needs validation. It is always important to evaluate the predictive power of a model using a held-out set (or better with cross-validation) prior to computing importances. Permutation importance does not reflect to the intrinsic predictive value of a feature by itself but how important this feature is for a particular model." (https://scikit-arn.org/stable/modules/permutation_importance.html).
5. Given the above points, the split to groups for discovering improvement in mature oocytes yield using percentages of follicles only in the 13-18 range is insufficient, the fact that it gave an increase does not necessarily mean its the best indicator, who is to say that using a range of 14-20 or even 18-22 wouldn't have given even better results? If the authors wish to give a global recommendation in which range most oocytes should be when giving trigger that is not depended on any model, they should test all possible ranges and show a comparison of them all. In that case, no model is even needed.
6. Information regarding relevant cycle parameters are missing. For example - were all cycles included or ICSI only cycles? How many cycles had agonist trigger only for oocyte maturation? Were these factors controlled in the regression analysis. These may impact the number of total and mature oocytes and possibly pregnancy rates.
7. It would be interesting to understand whether the results are generalized to all number of total follicles. For example If there are <10 follicles or 20-30 follicles (meaning possibly two cohorts of follicles), do outcomes improve for both groups when the majority of follicles are 13-18?
8. Progesterone rise can indeed decrease pregnancy and live birth rates but can be overcome by the freeze-all policy which

should be mentioned.

9. One of the major concerns relying on follicular size measurement during stimulation is its accuracy and high inter- and intra- observer variability, especially when the number of follicles increases. Recently, AI Algorithms have been applied to predict oocyte triggering in combination with 3-dimensional ultrasound (3DUS) imaging. I would suggest adding this option (Liang X, Fang J, Li H, Yang X, Ni D, Zeng F, et al. CR-UNET-Based Ultrasonic Follicle Monitoring to Reduce Diameter Variability and Generate Area Automatically as a Novel Biomarker for Follicular Maturity. *Ultrasound Med Biol* [Internet] 2020;46(11):3125–34. Available from: <http://www.ncbi.nlm.nih.gov/pubmed/32839052>).

Reviewer #2

(Remarks to the Author)

This interesting and innovative study utilizes machine learning approaches to investigate factors contributing to IVF outcomes, specifically mature oocytes subsequently retrieved and live birth rates, using data collected from 11 clinics in Europe. Furthermore, the authors employed XAI tools to gain a better understanding of the significance and contribution of these factors. The main finding indicated that follicles of sizes 13-18 mm on the day of trigger mostly contribute to the retrieval of mature oocytes and subsequent embryo development. In general, the data is of high quality and so are the statistical, the algorithmic approaches, and the evaluation procedures. Therefore, I believe the results and conclusions are robust and reliable.

Major comments:

1. Why did the authors use only one machine learning algorithm? Using additional algorithms may strengthen the conclusions. As the authors used a high amount of data, neural networks may be advisable to use.
2. Smoking (or exposure to smoking) is known to affect the hormonal system and the IVF outcomes. Despite its significance, this factor is neither addressed in the introduction nor investigated in this research. It should be addressed, at least in the introduction. If not addressed in the manuscript, the authors should at least explain why.
3. The data is based on clinics from two countries – did the authors compare the clinical measures between the two countries? It would be valuable to address any potential differences in clinical outcomes between the two countries to rule out any potential bias in the results.

Minor comments:

1. Page 4 line 61 – why was the threshold of 35 years old chosen? Did the researchers try other thresholds?
2. Page 4 figure 2d – what is the explanation to the drop in the normalized importance of follicles of size 16 mm?
3. Page 4 – line 66 – please correct the sentence grammar.
4. Table 1 – there is a discrepancy between how SD is described in the table's legend and how SD is presented in the table (with plus/minus sign, in parentheses?)
5. Table 2 – ranges of values are also required.
6. Page 6 – line 91 – 4 digits after the decimal point are not common, please use 2 or 3.
7. Page 6 – line 107 – did the authors calculate any correlation between mean follicle size and LBR? If so, what is the correlation coefficient and why isn't it mentioned in the methods section?
8. Figure 4b- the authors should consider using Kolmogorov-Smirnov test.
9. Page 7 – line 108 – consider move to discussion.
10. Page 12 – lines 273-278 and also line 288-291 are not clear, please rephrase or elaborate.

Version 1:

Reviewer comments:

Reviewer #1

(Remarks to the Author)

I appreciate the authors' efforts in addressing my comments.

However, I still have some concerns that are critical to the evaluation of the work:

1. Model Performance:

My main concern of this work is the low R^2 with a relatively high MAE.

- More information is needed to thoroughly analyze the results, particularly to understand the relatively low R^2 , which is strongly connected to the mean known MII and total eggs in the data. Please add these parameters.
- It would also be very helpful to include a plot of the predicted vs. known values (for each clinic or all of them together). This visualization will provide a more intuitive understanding of the model's prediction performance and reliability.

2. Statistics on the Data:

- Comprehensive statistics on the dataset are essential for assessing the context and reliability of the results. The authors should provide detailed information, including the mean and standard deviation (STD), as well as the range (minimum and maximum values) for all variables presented in Table 2.

These additional details will greatly enhance the clarity of the relatively low performance model presented in the paper.

Reviewer #2

(Remarks to the Author)

I reviewed the responses to my comments and the revised manuscript. The manuscript has significantly improved and all my comments were thoroughly addressed. Therefore, I recommend accepting the manuscript for publication.

Version 2:

Reviewer comments:

Reviewer #1

(Remarks to the Author)

I reviewed the responses to my comments and the revised manuscript. The manuscript has improved and my comments were addressed. Therefore, I recommend accepting the manuscript for publication.

REVIEWER COMMENTS

Reviewer #1 (Remarks to the Author):

The manuscript by Hanassab et al. used explainable artificial intelligence (XAI) to identify follicle sizes that contribute most to IVF cycle outcomes. For this end, the authors performed a multicenter study including 11 European countries with 19,082 patients. They concluded that follicles in the size range of 13-18 mm on the day of trigger contribute the most to the retrieval of mature oocytes and subsequent embryo development and that maximizing this proportion of follicles by the end of ovarian stimulation was associated with improved live birth rates. Furthermore, they suggested that larger mean follicle sizes, especially those >18 mm, were associated with progesterone elevation by the end of the ovarian stimulation with a possible negative impact on live birth rates.

The topic, which has been previously published, is interesting and may aid physicians in personalization and optimization of cycle outcomes. Its main advantages are the large sample size, its generality representing different clinics and practices and its explainability which may contribute to its acceptance in daily use. Although there are already several publications on trigger day algorithms, the study is novel in its approach. However, several comments and concerns need to be addressed by the authors.

Comments:

1. The authors trained 11 different models by repeatedly selecting 1 clinic to act as test set, and using the other 10 clinics to train the model. It is unclear if during hyper-parameter tuning the pipeline had access only to the 10 clinics data, or it was able to repeatedly test itself against the final left out clinic's data. A more elaborate explanation on the training, validation and testing methodology should be supplied.

Thank you for your positive appraisal and recognizing the value of our manuscript. We provide further clarification of the methodology used in the cross-validation pipeline as follows. At every cross-validation fold, the 11th fold is an independent test set. Within the 10 folds that remain, the 10th is a validation set used to tune hyperparameters and all the rest (comprising 9 folds) are used as a training set.

To mitigate information leakage across validation folds, we used "nested" cross-validation, specifically leave-one-clinic-out cross-validation (LOCO-CV), to carry out hyperparameter tuning as described in:

https://scikit-learn.org/stable/auto_examples/model_selection/plot_nested_cross_validation_iris.html.

We have now provided the code used with the submission to enable further inspection.

We have added to the following text to clarify this for the reader in the Methods:

"The mean absolute error (MAE) was optimized as an objective function using nested LOCO-CV with Bayesian optimization to tune relevant hyperparameters in the search space demonstrated in Supplementary Table 2. This procedure is where at every LOCO-CV fold, the eleventh fold (clinic) represents an independent test set. Within the other ten folds, the tenth is a validation set for tuning hyperparameters, and the remaining nine are used as a training set." Lines 262-265.

2. My major concern is the average performance of the article models in predicting M2 oocytes on day of trigger (MAE 3.6, R2 0.35) that underperforms compared to similar models in existing publications (Reuvenny - MAE 2.2, R2 0.69 ; Fanton – 2.87, R2 – 0.64), this weakens the results of the article, and should be discussed.

Thank you for requesting further clarification regarding the model performance reported in the present work in relation to previously reported models. Although the reported mean absolute error (MAE) in our model is higher than those reported in the studies by Reuvenny et al 2024 and Fanton et al 2022, we believe that this reflects the robust approach that we have chosen to use in the development protocol that should make our findings more generalizable and applicable to clinics across the world.

Firstly, we sourced data from 11 centers in two countries and employed a leave-one-clinic-out cross-validation (LOCO-CV) procedure that mitigates against overfitting, which should make the model more generalizable.

The second major reason for the lower error reported in the previous models is that they applied restrictions to the data to exclude atypical or extreme results (e.g. exclusion of patients with >30 oocytes retrieved in Reuvenny et al 2024) that would have resulted in a higher error and would have reduced the model performance towards the true performance. As we did not apply such restrictions to the source data, our estimates more closely represent the true error observed in real-world clinical practice.

We expand on these two primary reasons below:

1. **The development protocol:** the best-advised machine learning procedure when handling retrospective data from multiple clinics is 'internal-external validation', as described in Steyerberg & Harrell 2016, further endorsed by Collins et al 2024 for the appropriate evaluation of clinical prediction models. Since data in our study were sourced from 11 clinics across two countries and multiple cities, we implemented the leave-one-clinic-out cross-validation (LOCO-CV) procedure as the most robust approach.

It should be noted that data in Reuvenny et al 2024 was only sourced from one clinic, and in Fanton et al 2022 from only three clinics in one country, with a random split chosen across training, validation, and test sets in both cases. This therefore limits the methodology used in those studies to only internal validation, where it is "*strongly advise[d] against random split sample approaches in small development samples*" (Steyerberg & Harrell, 2016), especially when data from multiple sources are available (Collins et al 2024; Lasko et al, 2024; Wiens et al., 2019). Therefore, our validation procedure mitigates against the susceptibility to clinic-specific bias and resultant overfitting, providing a more transparent view of model transportability and generalizability. Hence, we were able to capture any variations in practice noted on a regional/national level, reported in both the mean absolute error (MAE) as the primary loss function across all clinics and, most importantly, included a standard deviation measure of the error across clinics. We believe this better represents the variation in performance in such predictive models across clinics, albeit findings on the most important follicle sizes were generally consistent across all 11 centers.

Additionally, we were careful to include only the first treatment cycle from treatment-naive patients in our model. Comparatively, the number of cycles per patient included is not clearly described in the Fanton and Reuvenny studies. Multiple cycles from a single patient will likely introduce longitudinal correlation across treatment attempts, and bias the true values of permutation importance for the input variables (Hu and Szymczak 2023). Additionally, the methods used in Fanton et al 2022 (i.e., linear regression) require independent input samples, which is likely not achieved if multiple cycles from the same patient are used in the absence of a hierarchical model.

We have now emphasized this point, and reference a relevant recent publication by Lasko et al 2024 in npj Digital Medicine in the Discussion on lines 172 - 178:

“Finally, we ensured only to include the first treatment cycle of each patient in this study for model development and validation (i.e., $n=19,082$ represents both the number of treatment-naive patients and cycles). This was to avoid auto-correlation between successive cycles of a single patient (e.g., a clinician’s decision-making is likely to be influenced by a previous treatment cycle) since longitudinal random effects across sequential cycles can influence the permutation importance (Hu and Szymczak 2023). Similarly, input parameters that present multi-collinearity (e.g., estradiol and follicle size (Abbara et al, 2019)) can also result in unreliable insights from permutation importance analysis.”

2. **Data curation:** Studies by both Fanton et al and Reuvenny et al curated their data to exclude atypical results. For instance, Fanton et al 2022 excluded cycles where the number of MII oocytes retrieved was higher than the follicles counted on the day of trigger as this could represent measurement error: *“Furthermore, cycles with apparent data entry errors were excluded, such as cycles in which the number of MII oocytes exceeded the number of oocytes retrieved.”* (Fanton et al 2022).

Similarly, Reuvenny et al 2024 also curated and restricted their source data to exclude any cycles with more than 30 MII oocytes retrieved, and introduced a second ‘refined’ model that had even stricter data curation: *“Further evaluation was performed on a refined subset (refined test), with excluded cycles that most likely represent erroneous data: (i) cycles detected by the algorithm as outliers, with statistically divergent results; and (ii) ‘aberrant cycles’, defined, based on expert opinion, as either an absolute difference of at least 10 between the number of follicles (>10 mm) and the number of retrieved oocytes, or a ratio of follicles to retrieved oocytes greater than 2 or less than 0.5 and an absolute difference of at least 5 between the two.”* (Reuvenny et al, 2024)

For interest, we have performed the same restriction techniques to compare model performance with and without such curation when trying to predict MII oocytes retrieved. When extending the curation to that described in Reuvenny et al 2024 as ‘aberrant cycles’ and removing any patient cycles with >30 MII oocytes retrieved (noted in 1,322 cycles in our instance), the model performance improves to MAE 2.92 (0.26), which is more comparable to the error reported in that study.

Additionally, Fanton et al 2022 imputed “missing” follicles on the basis of an unexplained oocyte number that was higher than follicle count, albeit reporting

minimal impact on model performance. Whilst curating the data is an understandable approach to remove potential measurement error and atypical results, this will inevitably artificially improve the performance of the models reported in these existing studies. Fanton et al 2022 used linear regression, however the ensemble-based method we used is better able to handle any unexpected noise due to measurement error (Liang et al, 2020) or biological phenomena (e.g., vanishing follicles; Dozortsev et al, 2022), and therefore we did not restrict our source data to avoid artificially improving our model error.

We highlight the above points regarding the potential downside of data-curation in lines 185 - 189 of the Discussion:

“Other studies have shown that follicle imputation has limited impact on enhancing model performance in this specific context (Fanton et al, 2022). Although excluding such treatment cycles could artificially improve model error, it is then not possible to build models that are more robust to such measurement error in future applications. Our approach utilizes an ensemble-based ML model with Bayesian optimization and can mitigate against the impact of data inconsistencies due to extremes of biological variation and/or measurement error (Liang et al, 2020).”

In light of the aforementioned reasons, we believe that the performance of our model is now better clarified as representing a more transparent approach with respect to model generalizability and transportability, which therefore should be more representative of the true performance in real-world clinical practice than those reported in previous studies.

References

Steyerberg, E.W. and Harrell Jr, F.E., 2016. Prediction models need appropriate internal, internal-external, and external validation. *Journal of clinical epidemiology*, 69, p.245.

Collins, G.S., Dhiman, P., Ma, J., Schlüssel, M.M., Archer, L., Van Calster, B., Harrell, F.E., Martin, G.P., Moons, K.G., Van Smeden, M. and Sperrin, M., 2024. Evaluation of clinical prediction models (part 1): from development to external validation. *bmj*, 384.

Lasko, T.A., Strobl, E.V. and Stead, W.W., 2024. Why do probabilistic clinical models fail to transport between sites. *npj Digital Medicine*, 7(1), p.53.

Wiens, J., Saria, S., Sendak, M., Ghassemi, M., Liu, V.X., Doshi-Velez, F., Jung, K., Heller, K., Kale, D., Saeed, M. and Ossorio, P.N., 2019. Do no harm: a roadmap for responsible machine learning for health care. *Nature medicine*, 25(9), pp.1337-1340.

Hu, J. and Szymczak, S., 2023. A review on longitudinal data analysis with random forest. *Briefings in Bioinformatics*, 24(2), p.bbaf002.

Abbara, A., Patel, A., Hunjan, T., Clarke, S.A., Chia, G., Eng, P.C., Phylactou, M., Cominos, A.N., Lavery, S., Trew, G.H. and Salim, R., 2019. FSH requirements for follicle growth during controlled ovarian stimulation. *Frontiers in endocrinology*, 10, p.579.

Liang, X., Fang, J., Li, H., Yang, X., Ni, D., Zeng, F. and Chen, Z., 2020. CR-Unet-based ultrasonic follicle monitoring to reduce diameter variability and generate area automatically as a novel biomarker for follicular maturity. *Ultrasound in Medicine & Biology*, 46(11), pp.3125-3134.

Dozortsev, D.I. and Diamond, M.P., 2022. Two peas from the same pod: vanishing follicles and postmature oocytes. *Fertility and sterility*, 117(1), pp.40-41.

3. The article focus on the effect of follicle size on the clinical outcome. It is known that many other parameters such as estradiol levels might be involved. It is not clear what was their impact on the model, which might weaken the effect of a model relying on the follicular size only. This effect should be reported and discussed.

We thank the reviewer for the helpful question. Follicles produce estradiol with larger follicles producing greater amounts of estradiol. Serum estradiol is therefore reflective of the follicle size profile and consequently, serum estradiol levels also correlate with the number of oocytes produced (Abbara et al, 2019). Although estradiol can influence serum LH, which in turn can affect the response to the trigger, it is not believed that the amount of estradiol produced is directly reflective of the ability of a follicle of a certain size to yield an oocyte *per se* if follicle size is accounted for.

Therefore, serum estradiol is not routinely measured in all centers to monitor cycles as it is not clear that it provides additional information over and above follicle size profiles. Furthermore, the European Society for Human Reproduction and Embryology Guideline for Ovarian Stimulation states: *“The addition of oestradiol measurements to ultrasound monitoring is probably not recommended.”* For that reason, serum estradiol was only available for a subset of patients (n=4,197 for cycles reporting oocytes and n=3,360 for cycles reporting mature oocytes) from the total number of cycles (n=19,082). For instance, the correlation coefficient for the relationship between the number of follicles sized 13-18mm and the number of oocytes retrieved is 0.684, whereas the correlation coefficient for serum estradiol on the day of trigger administration vs the number of oocytes retrieved is lower at 0.565.

For completeness, we have assessed the impact of including oestradiol in the model to predict the number of oocytes and mature oocytes, and we find that the addition of serum estradiol improved the model by only MAE 0.05 (MAE 4.30 reduces to 4.25) for the subset where estradiol was available (n=2068). This slightly improved further to MAE 4.19 when including the other treatment variables (as described previously in the manuscript) in the same patient subset. We have updated the relevant paragraph, now including estradiol, on lines 97-100 to read as follows:

“Incorporating other variables including: age, BMI, days of stimulation, the type of trigger, IVF protocol, and estradiol on the DoT administration (n=2,068) as separate input variables, only improved the MAE by 0.06 units (MAE: 3.54 ±0.36) and R2 by 0.01 (R2: 0.36 ±0.12). Overall, knowledge of follicle size on the DoT was the most important factor in estimating the number of MII oocytes retrieved.”

References:

Abbara, A., Patel, A., Hunjan, T., Clarke, S.A., Chia, G., Eng, P.C., Phylactou, M., Comninos, A.N., Lavery, S., Trew, G.H. and Salim, R., 2019. FSH requirements for follicle growth during controlled ovarian stimulation. *Frontiers in endocrinology*, 10, p.579.

ESHRE Guideline Group on Ovarian Stimulation, Bosch, E., Broer, S., Griesinger, G., Grynberg, M., Humaidan, P., Kolibianakis, E., Kunicki, M., La Marca, A., Lainas, G. and Le Clef, N., 2020. ESHRE guideline: ovarian stimulation for IVF/ICSI. *Human reproduction open*, 2020(2), p.hoaa009.

4. The article states that the high importance of some features (follicles 13-18) in their specific models is an indication that they are important in general for optimizing the number of oocytes retrieved, this needs validation. It is always important to evaluate the predictive power of a model using a held-out set (or better with cross-validation) prior to computing importances. Permutation importance does not reflect to the intrinsic predictive value of a feature by itself but how important this feature is for a particular model.” 5. Given the above points, the split to groups for discovering improvement in mature oocytes yield using percentages of follicles only in the 13-18 range is insufficient, the fact that it gave an increase does not necessarily mean it's the best indicator, who is to say that using a range of 14-20 or even 18-22 wouldn't have given even better results? If the authors wish to give a global recommendation in which range most oocytes should be when giving trigger that is not depended on any model, they should test all possible ranges and show a comparison of them all. In that case, no model is even needed.

The reviewer raises an excellent point in comments 4 and 5. We agree with the reviewer that our analysis seeks to address a specific and distinct question namely *‘which follicle sizes contribute relatively the most to the number of oocytes retrieved’*, rather than *‘which specific follicle size range is most predictive of the number of mature oocytes retrieved’*. We agree with the expert reviewer in stating that this nuanced distinction is important to recognise and to avoid misinterpretation of the results of the analysis presented.

Consequently, although we find that follicles of sizes 13-18mm had the greatest relative contribution to the number of mature oocytes retrieved, and whilst logically stimulation protocols should aim to maximise the proportion in this range, we do not intend to recommend this as a single follicle size range for use to most accurately predict the number of mature oocytes retrieved.

Although our results indicate that follicles of these sizes have the greatest relative contribution, our results also indicate that follicles of sizes outside of this range also contribute to the number of mature oocytes retrieved (albeit to a relatively lesser degree). Therefore, a single specific size range that includes follicles of sizes outside of this optimal range of most contributory follicles could yet correlate more closely with the number of mature oocytes subsequently retrieved in accordance with the reviewer's point.

Notably, our findings indicate that follicles of certain sizes contribute relatively more than others, and thus maximising the proportion of follicles within this range could optimise the number of oocytes that will be retrieved. This can be achieved by adjusting the stimulation protocol to ensure that the spread of follicles is tighter in order to maximise the proportion within the optimal range (Abbara et al 2018). To unequivocally confirm whether the anticipated benefits of such a strategy are borne out in practice would require further prospective evaluation. However, in the present study, we have conducted analyses to add credence to this hypothesis and to validate the importance of maximising the proportion of follicles within the most contributory range. We utilized the natural variation in response to ovarian stimulation to show that clinical outcomes including improvement in oocyte yield and live birth rates were improved in cycles where the proportion of follicles in this range were higher (Figures 5A and 4B).

In addition to the reviewer's point that the size range of most contributory follicle sizes is not necessarily the most predictive single size range (as the lesser contribution of other follicles outside of this range would not have been considered), a specific size range also assumes that all follicles within that range have equal/uniform contribution to the number of oocytes

retrieved. This is a cruder prediction than is possible using our ML model evaluating both the number and likely relative contribution of each follicle.

Consequently, in order to evaluate the predictive capability of our model for the number of mature oocytes retrieved, rather than a single specific follicle size range, we would recommend to include data on all follicles of different sizes and consider the relative likelihood of each follicle to yield a mature oocyte.

To investigate the predictive capability of this model, we have added the following analyses:

We first evaluated the predictive ability of specific single size ranges using univariable linear regression utilising leave-one-clinic-out cross-validation across the 11 clinics.

1. No consideration of follicle size: If we do not consider follicle size at all and include the total number of follicles on day of trigger regardless of size (i.e. all follicle sized 6-26mm), this results in a model with an MAE 3.92 (0.45) and R2 0.26 (0.15).
2. Which is the best performing single size range: Then, to identify which single follicle size range was most predictive of the number of oocytes retrieved, we compared all possible follicle size ranges and found that the number of follicles sized 12-23mm was most predictive (with respect to lowest MAE) of the number of mature oocytes retrieved (MAE 3.71 (0.23); R2 (0.31 (0.12))).
3. Predictive performance of our model incorporating the size of each individual follicle: However, our ML model, which considered the number of each individual follicle size as well as the relative contribution of a follicle of that size, had even greater predictive performance (MAE 3.60 (0.35); R2 (0.35 (0.13))). As mentioned in the answer to point 3 above, adding in a multitude of additional predictive variables (*age, BMI, days of stimulation, the type of trigger, IVF protocol, and estradiol on the DoT administration*) only had a small additional impact (MAE improved by 0.06), indicating the dominant impact of follicle size.

Model (MII Oocytes)	MAE	R2
All follicle sizes (6-26mm)	3.92 (0.45)	0.26 (0.15)
Best performing single follicle size range (12-23mm)	3.71 (0.37)	0.31 (0.12)
ML model utilising each individual follicle size	3.60 (0.35)	0.35 (0.13)

We have reviewed the manuscript to ensure that interpretation of the findings is clearly and accurately communicated throughout.

We have added the following paragraphs to the Methods and Results section:

“We evaluated the impact of different variables on the ability of the trained models to predict the number of MII oocytes retrieved. Whilst follicles of 13-18mm had the highest relative contribution to the number of mature oocytes retrieved, this does not imply that other follicles

were not also contributory, albeit to a relatively lesser extent. If we do not consider follicle size at all and include the total number of follicles on the DoT regardless of size (i.e., all follicle sized 6-26mm), the resulting model had an MAE 3.92 ± 0.45 and $R^2 0.26 \pm 0.15$. If considering a single follicle size range, the number of follicles sized 12-23mm was most predictive (with respect to the lowest MAE) of the number of mature oocytes retrieved (MAE 3.71 ± 0.23 ; $R^2 0.31 \pm 0.12$). However, our ML model, which considered the count of each individual follicle size, as well as the relative contribution of a follicle of that size, had even greater predictive performance (MAE 3.60 (0.35); $R^2 (0.35 (0.13))$ (Table 1).” Lines 87-95

“Separately, to identify which single follicle range was most predictive of mature oocytes, we evaluated the predictive ability of specific size ranges using univariable linear regression with LOCO-CV across the eleven clinics ($n=14,140$). Then, to identify which range was most predictive of the number of mature oocytes retrieved, we compared all possible follicle size ranges using the same method, optimized for MAE as the loss function.” Lines 298-301

We have also amended the Discussion as follows:

“Our data suggest that a novel approach to deciding when to administer the trigger of oocyte maturation could be based on the proportion of intermediary-sized follicles (e.g., 13-18mm) rather than the traditional threshold-based approach assessing when 2-3 lead follicles reach 17 or 18mm in size.” Lines 191-193

“It should be noted that although a range of follicle sizes that contribute relatively more than others varied marginally depending on the patient stratifications considered (Fig. 2). Ultimately, an ML model that considers individual follicle sizes and their relative contributions, in addition to patient characteristics, could be harnessed as part of a clinical decision support system (Hanassab et al 2024; Letterie et al 2020).” Lines 199-202

“In conclusion, we establish that intermediary-sized follicles on the day of trigger contribute the most to the retrieval of mature oocytes and subsequent embryo development. Utilizing the sizes of all follicles, rather than just the size of only the lead follicles, could offer a target for OS protocols and inform the timing of trigger administration to optimize clinical outcomes.” Lines 203-205

We agree with the expert reviewer’s other point that cross-validation is the best approach rather than using a hold-out set, and hence why we have used ‘internal-external validation’ in this analysis. This approach is recommended by Steyerberg & Harrell 2016, and further promoted by Collins et al 2024 in the BMJ for model development using datasets that include data from multiple clinics.

We also agree with the reviewer that computing permutation importances should occur after model evaluation. We would like to affirm that even though the permutation importance results are presented before the subsection on model performance in the manuscript, these were not computationally executed in this ordering. Permutation importance was calculated on the eleventh test set at the end of each cross-validation fold (as confirmed in the code provided).

We have now clarified this procedural ordering in the subsection ‘Identifying important follicle sizes’ on lines 285-286:

“Only once each model was trained and validated (using up to ten folds in total), was the mean and standard deviation determined across five runs of the permutation importance of features using the eleventh independent test set in the LOCO-CV protocol (Breiman 2001).”

References

- Steyerberg, E.W. and Harrell Jr, F.E., 2016. Prediction models need appropriate internal, internal-external, and external validation. *Journal of clinical epidemiology*, 69, p.245.
- Collins, G.S., Dhiman, P., Ma, J., Schlüssel, M.M., Archer, L., Van Calster, B., Harrell, F.E., Martin, G.P., Moons, K.G., Van Smeden, M. and Sperrin, M., 2024. Evaluation of clinical prediction models (part 1): from development to external validation. *bmj*, 384.
- Abbara, A., Vuong, L.N., Ho, V.N., Clarke, S.A., Jeffers, L., Comninou, A.N., Salim, R., Ho, T.M., Kelsey, T.W., Trew, G.H. and Humaidan, P., 2018. Follicle size on day of trigger most likely to yield a mature oocyte. *Frontiers in endocrinology*, 9, p.193.
- Hanassab, S., Abbara, A., Yeung, A.C., Voliotis, M., Tsaneva-Atanasova, K., Kelsey, T.W., Trew, G.H., Nelson, S.M., Heinis, T. and Dhillon, W.S., 2024. The prospect of artificial intelligence to personalize assisted reproductive technology. *npj Digital Medicine*, 7(1), p.55.
- Letterie, G. and Mac Donald, A., 2020. Artificial intelligence in in vitro fertilization: a computer decision support system for day-to-day management of ovarian stimulation during in vitro fertilization. *Fertility and Sterility*, 114(5), pp.1026-1031.
- Breiman, L., 2001. Random forests. *Machine learning*, 45, pp.5-32.

6. Information regarding relevant cycle parameters are missing. For example - were all cycles included or ICSI only cycles? How many cycles had agonist trigger only for oocyte maturation? Were these factors controlled in the regression analysis. These may impact the number of total and mature oocytes and possibly pregnancy rates.

We thank the reviewer for requesting additional information on cycle characteristics. We have included both IVF (n=19,082) cycles of which 14,140 had fertilisation using ICSI, as noted in the Methods subsection ‘In vitro fertilization protocol’:

“This was a retrospective cohort study analyzing follicle and oocyte data from IVF or ICSI cycles. The objective was to identify the follicle sizes on the DoT that are most likely to yield mature oocytes and therefore provide a target for ovarian stimulation and expected oocyte number.” Lines 226-228

Therefore, we have replaced phrases describing “maturity-graded” patients to clarify that these denote ICSI treatment cycles where maturity could be graded throughout the manuscript as well as in Table 1. For further clarifications, we have now also noted the number of GnRH agonist and GnRH antagonist suppressant protocol cycles in the “In vitro fertilization protocol” subsection on lines 235 -236 :

“Patients underwent a suppressant protocol to prevent premature ovulation using either a gonadotropin-releasing hormone (GnRH) agonist (‘long’ protocol; n=6,990) or antagonist (‘short’ protocol; n=7,408) co-treated protocol.”

With respect to the trigger administered, we considered the impact of this both in predicting mature oocytes, as well as live birth. We ensured consistency in the trigger administered (i.e., hCG trigger only) in the analysis of Figure 2c and 2d when comparing treatment protocols (i.e., long versus short). We did include trigger type as an input variable when using a multivariable model to predict mature oocytes alongside other variables such as age, BMI, or total follicle count (see lines 95-100), however, its impact was marginal compared to information on follicle sizes (model improved at most by MAE 0.06). As a sensitivity analysis we excluded patients who received a GnRH agonist trigger (n=1,181) in a new LOCO-CV procedure model, however, this made no difference to the most contributory follicles for mature oocytes (i.e., remained 13-18mm above 50% relative importance).

To confirm, we do already adjust for the oocyte maturation trigger administered (OR: 1.35 (1.11 - 1.64) for the hCG trigger; p=0.003) in the multivariable logistic regression analysis for live birth as described in the Methods subsection 'Associations with live birth rates':

“To determine any associations between the proposed follicle size and live birth rate (LBR), we used logistic regression on all data with LBR recorded (n=9,843) and input variables including the percentage of follicles in the proposed follicle size range, total follicle count on the DoT, age at the time of treatment, and type of trigger administered (hCG or GnRH agonist).” Lines 315-317

7. It would be interesting to understand whether the results are generalized to all number of total follicles. For example If there are <10 follicles or 20-30 follicles (meaning possibly two cohorts of follicles), do outcomes improve for both groups when the majority of follicles are 13-18?

Thank you for asking this interesting question regarding whether maximising the proportion of follicles within the optimal size range improves clinical outcomes in patients with different total follicle numbers.

With respect to mature oocyte yield (mature oocytes as a proportion of total follicle count), stratifying patients into subgroups of total follicle number on the day of trigger, i.e., <10, 10-19, or 20+ follicles, did not impact the trend for the relationship between mature oocyte yield and the proportion of follicles within the most contributory 13-18mm size range.

We had already included the total follicle number as a primary input variable in the logistic regression model for live birth (Figure 5a). To specifically observe the impact of the follicle count on the effect of the target 13-18mm range on live birth as per the comment, we have now investigated introducing an interaction effect between total follicle count and the percentage of follicle sizes in the 13-18mm range for the logistic regression model (as described in lines 315-317). The interaction term was not statistically significant ($p=0.285$), indicating that the effect of the percentage of follicles in the target range does not depend on the total follicle count.

In both cases, we conclude that maximising the percentage of follicles in the optimal intermediary-sized range improved clinical outcomes in patients with different follicle numbers.

8. Progesterone rise can indeed decrease pregnancy and live birth rates but can be overcome by the freeze-all policy which should be mentioned.

We agree with the reviewer that premature progesterone elevation at trigger day can lead to abnormal advancement of endometrial maturation and thus can be detrimental to pregnancy rates in the context of a fresh transfer. However, a freeze-all can mitigate the risk of reduced implantation at the expense of increasing the time-to-pregnancy and the risk of perinatal complications, e.g. large-for-gestational-age babies (Garg et al 2023). Our data reveal that the strategy for ovarian stimulation could be modified in the context of whether a fresh transfer is to be conducted.

We have clarified this distinction between fresh and frozen policies in the subsection 'In vitro fertilization protocol' on lines 229-234:

"Patients at elevated risk of ovarian hyperstimulation syndrome (OHSS) or who are noted to have premature progesterone elevation are often advised to have their embryos cryopreserved pending a frozen embryo transfer (called 'freeze-all') (Abbara et al, 2018). A freeze-all strategy can mitigate the risk of reduced implantation due to premature progesterone elevation, albeit at the expense of increasing the time-to-pregnancy and the risk of perinatal complications, e.g., large-for-gestational-age babies (Garg et al 2023). For analysis of live birth in this study, women underwent either IVF or ICSI with fresh embryo transfer and had their final ultrasound scan to assess follicle size on the DoT administration."

References:

Abbara, A., Vuong, L.N., Ho, V.N., Clarke, S.A., Jeffers, L., Comminos, A.N., Salim, R., Ho, T.M., Kelsey, T.W., Trew, G.H. and Humaidan, P., 2018. Follicle size on day of trigger most likely to yield a mature oocyte. *Frontiers in endocrinology*, 9, p.193.

Garg, A., Zielinska, A.P., Yeung, A.C., Abdelmalak, R., Chen, R., Hossain, A., Israni, A., Nelson, S.M., Babwah, A.V., Dhillo, W.S. and Abbara, A., 2024. Luteal phase support in assisted reproductive technology. *Nature Reviews Endocrinology*, 20(3), pp.149-167.

9. One of the major concerns relying on follicular size measurement during stimulation is its accuracy and high inter- and intra- observer variability, especially when the number of follicles increases. Recently, AI Algorithms have been applied to predict oocyte triggering in combination with 3-dimensional ultrasound (3DUS)

imaging. I would suggest adding this option (Liang X, Fang J, Li H, Yang X, Ni D, Zeng F, et al. CR-Unet-Based Ultrasonic Follicle Monitoring to Reduce Diameter Variability and Generate Area Automatically as a Novel Biomarker for Follicular Maturity. *Ultrasound Med Biol* [Internet] 2020;46(11):3125–34. Available from: <http://www.ncbi.nlm.nih.gov/pubmed/32839052>).

Thank you for recommending this valuable research regarding automated ultrasound scanning of follicles. In line with our motivation for improved accuracy and reliability of follicle measurements described in lines 179-180, we have now included this reference at the end of lines 179, 187, and 190:

“Variability in follicle size measurements, both within and between observers, has been a documented challenge in ART (Liang et al, 2020)”

“Our approach utilizes an ensemble-based ML model with Bayesian optimization and can mitigate against the impact of data inconsistencies due to extremes of biological variation and/or measurement error (Liang et al, 2020)”

“The need for more objective ultrasound scanning methods, potentially through the integration of automated algorithms, may further improve the accuracy and reliability of follicle measurements and associated algorithms in ART (Liang et al, 2020).”

Reference:

Liang, X., Fang, J., Li, H., Yang, X., Ni, D., Zeng, F. and Chen, Z., 2020. CR-Unet-based ultrasonic follicle monitoring to reduce diameter variability and generate area automatically as a novel biomarker for follicular maturity. *Ultrasound in Medicine & Biology*, 46(11), pp.3125-3134.

Reviewer #2 (Remarks to the Author):

This interesting and innovative study utilizes machine learning approaches to investigate factors contributing to IVF outcomes, specifically mature oocytes subsequently retrieved and live birth rates, using data collected from 11 clinics in Europe. Furthermore, the authors employed XAI tools to gain a better understanding of the significance and contribution of these factors. The main finding indicated that follicles of sizes 13-18 mm on the day of trigger mostly contribute to the retrieval of mature oocytes and subsequent embryo development. In general, the data is of high quality and so are the statistical, the algorithmic approaches, and the evaluation procedures. Therefore, I believe the results and conclusions are robust and reliable.

Major comments:

1. Why did the authors use only one machine learning algorithm? Using additional algorithms may strengthen the conclusions. As the authors used a high amount of data, neural networks may be advisable to use.

We thank the reviewer for their positive appraisal of our manuscript. When choosing an appropriate algorithm, we sought a method that could handle non-linear effects, had an intuitive architecture, and most importantly, could be explainable. We considered other ML algorithms, both simpler (e.g., linear regression) and more complex (e.g., stacked meta-algorithms like AutoGluon), however, performance was not significantly improved and therefore the histogram-based gradient boosting regression model was deemed most suitable to answer this research question.

We thank the reviewer for specifically suggesting an artificial neural network as a potential modeling candidate. In light of this, we have implemented a multilayer perceptron regression model ('MLPRegressor') aimed at the primary MII (mature) oocytes outcome (n=14,140 patients). This was achieved with appropriate hyperparameter tuning using the same Bayesian optimization pipeline with leave-one-clinic-out cross-validation (LOCO-CV) as outlined in the Methods. In this case, we tuned for the number of hidden layers (2-20), the learning rate (0.0001-0.1), and alpha value (0.0001-0.1).

(https://scikit-learn.org/stable/modules/generated/sklearn.neural_network.MLPRegressor.html)

The average performances are outlined in the table below:

Metric	Mean (SD)
Mean Absolute Error (MAE)	3.85 (0.53)
Coefficient of Determination (R2)	0.20 (0.19)
Median Absolute Error (MedAE)	2.69 (0.36)
Root mean squared Error (RMSE)	5.62 (1.06)
Maximum Error	43.7 (34.0)

The comparative primary model described in the paper for MII (mature) oocytes achieved a superior MAE of 3.60 (0.35). Nonetheless, the model identified 14-18mm follicles

as the most important (similar to our other results). We assume that despite the relatively high number of samples in the dataset, the complexity of the underlying data is not sufficient for a neural network architecture to perform with superiority, and thus likely overfitting the data.

This new analysis is now included in the Results and Methods:

“In contrast, the multilayer perceptron model for predicting MII oocytes presented a higher MAE 3.85 (0.53), identifying 14-18mm follicles as the most important.” Lines 79-80

“Similarly, to observe any notable impact in the trade-off between model complexity and explainability, we modelled the primary outcome of MII oocytes (n=14,140 patients) using a multilayer perceptron model (a shallow artificial neural network) and reported its MAE.” Lines 276-278

2. Smoking (or exposure to smoking) is known to affect the hormonal system and the IVF outcomes. Despite its significance, this factor is neither addressed in the introduction nor investigated in this research. It should be addressed, at least in the introduction. If not addressed in the manuscript, the authors should at least explain why.

We thank the reviewer for highlighting this important point regarding smoking. As the reviewer has rightly recognized, smoking is known to have an adverse impact on IVF outcomes, as noted by the NICE guidelines.

In the UK, there is a policy adopted by the integrated care boards (ICBs) of the associated clinics whereby state-funded IVF cycles require both partners to be non-smokers for at least 3 months prior to the first treatment attempt. This relies on self-reporting from patients and may be subject to verification by cotinine breath testing. Thus, the vast majority of patients were likely to be non-smokers but unfortunately only ~1% of patients had data regarding smoking recorded, and therefore we were not able to consider this formally as part of the main analyses.

We have added the following sentence to the Discussion as a limitation to the analysis:

“Likewise, although most patients are required to be non-smokers to access state-funded care, there was insufficient recording of smoking status to formally assess any impact of smoking.” Lines 152-153

Reference:

<https://www.nice.org.uk/guidance/cg156/ifp/chapter/in-vitro-fertilisation>

3. The data is based on clinics from two countries – did the authors compare the clinical measures between the two countries? It would be valuable to address any potential differences in clinical outcomes between the two countries to rule out any potential bias in the results.

We agree with the reviewer that there may be differences in clinical outcomes with respect to different clinics as well as in different countries. This could reflect both differences in the population served including environmental and socio-economic factors, as well as differences in treatment protocols and laboratory facilities/procedures. Therefore, we have

adopted an approach that could take account of such differences between centers, namely the leave-one-clinic-out cross-validation (LOCO-CV) protocol, whereby differences in model performance can be highlighted, and we are able to report a mean and standard deviation across all clinics involved. This approach should make our findings more generalizable to data from new unseen clinics.

To confirm that the main findings are not impacted by geographical factors, we performed a sensitivity analysis including UK clinics only (n=11,960 patients from 9 clinics). The most contributory follicle sizes remained similar (see below) with a comparable MAE at 3.66 (0.34), as data reported in Table 2 when all clinics were included (from Fig. 2b below).

Minor comments:

1. Page 4 line 61 – why was the threshold of 35 years old chosen? Did the researchers try other thresholds?

Thank you for this comment. The threshold of 35 years is a common threshold often used in fertility research studies to denote relatively young women with predicted good responses to ovarian stimulation. This is traditionally used since age-related ovarian reserve decline accelerates starkly after the age of 35 years (Owen and Sparzak, 2022). The

reported national statistics of the United States (CDC) and United Kingdom (HFEA) both report a mean age of 35 years, which is also reflected in our data (mean age of 34.61 years). Furthermore, the American Society for Reproductive Medicine (ASRM) defines “infertility” specifically for two age brackets, those 35 years and older, or below:

“In patients having regular, unprotected intercourse and without any known etiology for either partner suggestive of impaired reproductive ability, evaluation should be initiated at 12 months when the female partner is under 35 years of age and at 6 months when the female partner is 35 years of age or older.” (ASRM)

We used this threshold as part of an exploratory analysis to assess whether predicted good responders (generally younger women) would exhibit a different follicle importance profile to older patients, however, the optimal follicle sizes appeared to be similar in both groups.

As noted in the Discussion on lines 150-152:

“...it has been suggested that older patients with diminished ovarian reserve may benefit from earlier trigger administration in modified natural cycles (Lawrenz et al 2024). To date, the impact of age on trigger timing in IVF cycles remains uncertain.”

References:

Owen, A. and Sparzak, P.B., 2022. Age related fertility decline.

ASRM definition of infertility:

https://www.asrm.org/globalassets/_asrm/practice-guidance/practice-guidelines/pdf/definition-of-infertility.pdf

Lawrenz, B., Kalafat, E., Ata, B., Melado, L., Del Gallego, R., Elkhatib, I. and Fatemi, H., 2024. Do women with severely diminished ovarian reserve undergoing modified natural cycles benefit from earlier trigger at smaller follicle size?. *Ultrasound in Obstetrics & Gynecology*.

2. Page 4 figure 2d – what is the explanation to the drop in the normalized importance of follicles of size 16 mm?

Thank you for this interesting observation in this subanalysis. Although there is a drop at 16mm, it remains in the top 50% of permutation importance consistent with adjacent follicle sizes. The drop most likely represents a chance finding as there is no biological plausibility for a drop in normalized importance at this size, and a similar observation was not seen in other analyses with larger samples. However, we have investigated fully to ascertain whether an alternative explanation could be identified.

In the context of the analysis presented in the paper, the main potential factors that could have influenced permutation importance estimates are a) correlations between variables, b) operator variability, and c) sample size and statistical variability. In light of this comment, the following additional analyses were carried out on the data used to create Figure 2D to investigate the dip in permutation importance of follicles of size 16mm.

a) We double-checked the correlations and collinearity in the data used for producing Figure 2D. There were no strong correlations between variables. We performed an additional linear regression analysis with logged mature oocytes as the dependent variable

to investigate collinearity using the variance inflation factor (VIF). All VIF values were below 1.67, which indicates that there are no concerns of collinearity.

b) In the context of this analysis, we have assessed whether there was evidence of operator variability related to scanning, reading, and the recording of follicle size and oocyte counts. We performed analysis of influential points using linear regression with logged mature oocytes as the dependent variable and looked at the "dfbetas". We noted that a small number of patients had a large negative effect on the regression coefficient estimates of follicles of size 16mm.

c) For the analysis in Figure 2D, one of the clinics had only 27 samples, which, for the most part, had a negative effect on the estimated permutation importance measures.

Overall, as there is no biological plausibility for this drop at 16mm and it was only seen in this one subanalysis, we conclude that the dip at 16mm follicles in Figure 2D is most likely a chance finding that is unlikely to represent a meaningful insight.

3. Page 4 – line 66 – please correct the sentence grammar.

We thank the reviewer for noting this. We have reformulated this sentence to make it clearer as follows:

“Follicles sized 14-20mm contributed most to MII oocytes in long protocol cycles (Fig.2(c)), whereas follicles sized 12-19mm were most important in short protocol cycles (Fig.2(d)).” Lines 65-66

4. Table 1 – there is a discrepancy between how SD is described in the table’s legend and how SD is presented in the table (with plus/minus sign, in parentheses?)

We thank the reviewer for noting this. The caption for Table 1 has been corrected to reflect the current syntax in Table 1, i.e. mean (SD).

5. Table 2 – ranges of values are also required.

Thank you for highlighting this point. We have re-populated Table 2 to include a measure of spread such as Standard Deviation or Interquartile Range as appropriate depending on the distribution. For age, which was normally distributed, we have used the mean and standard deviation. For body mass index (BMI), which was right-skewed, we have reported the median and lower (LQ) to upper quartile (UQ) ranges. Similarly, for all count data and clinical rates we have reported the median (LQ-UQ). Live birth rate is reported with its mean and 95% confidence interval.

Table 2 has now been updated as follows:

Patient detail	Measure	Patients in calculation
Age at treatment	34.61 (4.50)	19,080
Body mass index	24.17 (21.72 – 27.51)	10,965
Antral follicle count	15.00 (8.00 – 23.00)	9,620
Days to trigger	11.00 (10.00 – 12.00)	19,080
Follicles >10mm on DoT	11.00 (8.00 – 15.00)	19,082
No. oocytes collected	11.00 (7.00 – 16.00)	19,082
No. MII oocytes	8.00 (5.00 – 13.00)	14,140
No. 2PN zygotes	6.00 (3.00 – 10.00)	17,822
No. HQ blastocysts	2.00 (1.00 – 4.00)	17,488
Mature oocyte yield	0.59 (0.40 – 0.83)	14,140
Maturation rate	0.80 (0.67 – 0.92)	14,140
Fertilization rate	0.73 (0.56 – 0.86)	13,402
Blastulation rate	0.50 (0.29 – 0.67)	16,513
Live birth rate	30.48% (29.68% – 31.29%)	12,724

Table 2. Patient demographics and treatment cycle information from the total population of 19,082 participants. Data are reported as mean (standard deviation), median (lower quartile – upper quartile), or percentage (95% confidence interval). Mature oocyte yield is defined as the number of metaphase-II (MII) mature oocytes collected as a proportion of total follicle count on the day of trigger (DoT) administration. No.: number of; 2PN: two-pronuclear; HQ: high-quality.

6. Page 6 – line 91 – 4 digits after the decimal point are not common, please use 2 or 3.

Thank you for noting this and we have now revised the p-value to 3 decimal places on line 100 as: p=0.229.

7. Page 6 – line 107 – did the authors calculate any correlation between mean follicle size and LBR? If so, what is the correlation coefficient and why isn't it mentioned in the methods section?

Thank you for noting that although statistical significance was presented, the odds ratio (effect size) of this effect on live birth rate (LBR) was not detailed in the manuscript. We have now included the odds ratio (0.95) alongside its respective statistical significance (p=0.001). For further affirmation and interest to the reviewer, we have calculated the point biserial correlation between mean follicle size and live birth rate: -0.08 (-0.093 to -0.060).

This shows how the mean follicle size negatively impacts live birth rate as the mean size changes by 1mm (see lines 117-118):

“We next examined whether the mean follicle size impacted on LBR, and found a negative association (OR: 0.95 (0.93 - 0.98) per 1mm change; p=0.001).”

8. Figure 4b- the authors should consider using Kolmogorov-Smirnov test.

We thank you for suggesting this alternative non-parametric test for use in the analysis shown in Fig. 4b. The Mann-Whitney U test is a widely used non-parametric test for testing differences between the locations of two samples, assuming that the shape of the distributions is the same. The Kolmogorov-Smirnov test is advised against when handling count data and is *“valid only for continuous distributions”*.

<https://docs.scipy.org/doc/scipy/reference/generated/scipy.stats.kstest.html>.

This is discussed more extensively by Drew et al, 2000. We therefore decided to use the Mann-Whitney U test in this instance due to the nature of the data.

Reference:

Drew, J.H., Glen, A.G. and Leemis, L.M., 2000. Computing the cumulative distribution function of the Kolmogorov–Smirnov statistic. *Computational statistics & data analysis*, 34(1), pp.1-15.

9. Page 7 – line 108 – consider move to discussion.

We thank the reviewer for this suggestion and have moved this sentence from the Results section to the Discussion on lines 128-130:

“Furthermore, extending the duration of OS resulted in a greater number of larger follicles (>18mm) on the DoT that not only contributed less to the yield of mature oocytes but also resulted in premature progesterone elevation with a consequent negative impact on live birth (Venetis et al, 2013; Bosch et al, 2010), possibly due to its adverse effect on the endometrial stage (Garg et al, 2023).”

References:

Venetis, C.A., Kolibianakis, E.M., Bosdou, J.K. and Tarlatzis, B.C., 2013. Progesterone elevation and probability of pregnancy after IVF: a systematic review and meta-analysis of over 60 000 cycles. *Human reproduction update*, 19(5), pp.433-457.

Bosch, E., Labarta, E., Crespo, J., Simon, C., Remohi, J., Jenkins, J. and Pellicer, A., 2010. Circulating progesterone levels and ongoing pregnancy rates in controlled ovarian stimulation cycles for in vitro fertilization: analysis of over 4000 cycles. *Human reproduction*, 25(8), pp.2092-2100.

Garg, A., Zielinska, A.P., Yeung, A.C., Abdelmalak, R., Chen, R., Hossain, A., Israni, A., Nelson, S.M., Babwah, A.V., Dhillo, W.S. and Abbara, A., 2024. Luteal phase support in assisted reproductive technology. *Nature Reviews Endocrinology*, 20(3), pp.149-167.

10. Page 12 – lines 273-278 and also line 288-291 are not clear, please rephrase or elaborate.

We apologise for the lack of clarity. We have reformulated these two sections to make them clearer to follow, now on lines 302-313 and 321-324:

“We considered the cohort of patients who had ICSI (n=14,140) (where the number of MII oocytes was recorded) to compare the threshold-based criteria currently used in clinical practice, and a proposed approach based on maximizing the proportion of follicles within the optimal size range. For each of the four variations of the typically used threshold-based criteria to determine the DoT administration (i.e. two or three follicles greater than size 17 or 18mm), we assessed whether each patient cycle had fulfilled this criterion or not, and grouped them accordingly. We compared the relative difference in medians of mature oocyte yield (number of mature oocytes as a proportion of total follicle count on DoT) in these two groups of patients and compared the subgroups using the Mann-Whitney U test (Fig. 4a).

For the range-based criteria, we ran the same analysis at different minimum cut-offs of the percentage of follicles within that follicle size range on the DoT (e.g., ≥5%, ≥10%, and

so on), to examine any improvement in mature oocyte yield when maximizing the follicle sizes in this range (Fig. 4b). All statistical comparisons were carried out using the Mann-Whitney U test.”

“Further, we plotted the mean and 95% CI of the mature oocyte yield (n=646) and LBR (n=427) according to serum progesterone levels (nmol/L) on the DoT in 1nmol/L increments (Fig. 5c). We compared serum progesterone on the DoT (n=994) according to the number of follicles sized larger than the proposed optimal follicle size range (Fig. 5d). Progesterone levels were compared to those with less than two larger follicles using the Dunnett’s multiple comparison test.”

REVIEWER COMMENTS

Reviewer #1 (Remarks to the Author):

I appreciate the authors' efforts in addressing my comments.

However, I still have some concerns that are critical to the evaluation of the work:

1. Model Performance:

My main concern of this work is the low R² with a relatively high MAE.

- More information is needed to thoroughly analyze the results, particularly to understand the relatively low R², which is strongly connected to the mean known MII and total eggs in the data. Please add these parameters.

We thank the reviewer for the recognition of our efforts to further improve the manuscript. We appreciate the importance of delving deeper into explaining the model performance reported in Table 1, which is an average of performances across the 11 clinic folds in the leave-one-clinic-out cross-validation (LOCO-CV) procedure.

We believe that there are three core elements to expand on to fully investigate this:

1. Unbiased estimation of generalization error

It is important to stress the value of using 11 clinics as part of the present analysis, where the LOCO-CV procedure allows us to observe and validate predictive performance as a form of internal-external validation (Steyerberg and Harrell, 2016; Collins et al, 2024). This procedure leads to more robust and realistic estimates of predictive performance in diverse real-life scenarios such as different clinical settings and practices, as well as in different patient populations.

The standard deviations reported in Table 1 indicate that there is variation in performance between clinics. When internal-external validation is not used, there is a risk of overfitting which could be the case in previous studies where only one clinic (Reuveny et al, 2024) or three clinics (Fanton et al, 2022) were included using a random-split procedure, which is “*strongly advise[d] against ... in small development samples*” (Steyerberg and Harrell, 2016).

Therefore, a comparison of the model performance in the current study using internal-externally validated models with the only internally validated models based on curated data in the existing literature should be considered with this context in mind.

2. Investigating the drivers of high MAE in the current study

We can see in **Figure R1** that a few very high absolute errors are responsible for the higher mean absolute error. These prediction errors could be due to data collection or recording noise, the presence of natural outliers, or poor predictive performance of the models. **Figure R1** shows that the median absolute error (MedAE) (as reported

in **Table 1**) remains consistently lower across all clinics. Therefore, the first two reasons for high absolute errors seem more plausible.

Figure R1: Absolute prediction error in MII oocytes ($n=14,140$ patients) with the mean (in red), median (black line), interquartile range (black box), and outliers (black data points) defined as a point lying more than 1.5 times the interquartile range above the upper quartile.

Actual prediction errors are shown in **Figure R2**, where it can be seen that larger errors are driven by underprediction more so than overprediction.

For example:

- 1) In clinic A, 131 patients were reported to have 0 MII oocytes, which is the highest proportion across all clinics. Of these, 128 had at least 1 2PN zygote reported, which implies that these MII oocyte records data are likely

misrecorded. All these patients were predicted to have >2 MII oocytes collected, which likely overestimated the reported error.

- 2) Similarly, some patients had a particularly high number of MII oocytes collected that were underestimated by the models, which in turn also resulted in large errors. For example, one patient had 27 MII oocytes collected yet only 8 follicles were recorded on trigger day (a limitation likely reflecting operator variability in the recording of follicles as noted in the Discussion on lines 184-195).

Figure R2: Actual prediction error in MII oocytes (n=14,140 patients). Negative values correspond to overprediction, positive values correspond to underprediction.

As discussed in the previous rebuttal, Reuveny et al applied a curation procedure to remove all cycles with >30 oocytes from their single-clinic dataset as well as outliers and ‘*aberrant cycles defined based on expert opinion*’. Similarly, Fanton et al also excluded cycles with “*apparent data entry errors*”. We previously showed that carrying out similar data curation procedures also improves the model performance across the 11 clinics in our analysis leading to an average MAE of 2.92.

In view of the above, as a sensitivity analysis, we restricted the dataset to include only patients with 1-30 MII oocytes and where the number of MII oocytes collected was less than or equal to the total follicle count on trigger day (n=11,819 patients). This improved the average MAE to 2.54 (0.45) and R^2 0.49 (0.06) (as compared to 3.60 (0.35) and R^2 0.35 (0.13)). This confirms our understanding of how data extremities have impacted the MAE in our dataset. We have updated the Methods and Results to note this and included **Supplementary Figure 1** to present these results.

We have added the following sentence to the Methods on lines 276-278:

“Furthermore, to investigate the impact of outliers and potential aberrant data, we restricted the dataset to cycles with 1-30 MII oocytes retrieved, and where the number of follicles on the DoT was at least the number of MII oocytes retrieved (n=11,819 patients).”

and on lines 82-84 in the Results:

“When excluding potential aberrant data for this ICSI population in a separate model predicting MII oocytes (n=11,819), the average MAE improved to 2.54 (0.45) and R^2 to 0.49 (0.06).”

3. Low R^2 scores are generally driven by an uncharacteristically low association between follicle sizes and MII oocytes.

To understand why certain clinics had lower R^2 scores compared with others, we considered the Pearson correlation coefficient between the total follicle size count on the day of trigger and the number of MII oocytes collected in each of the 11 clinics separately (**Table R1**). Generally, we would expect a strong correlation between these two measures. However, the two clinics that had low R^2 scores for model performance in the LOCO-CV (i.e., clinics B and D) also had the lowest Spearman’s correlation coefficients, 0.44 and 0.40, respectively.

We also considered the multivariable R^2 by regressing the number of MII oocytes against individual follicle size counts. Again, clinic D had the lowest R^2 at 0.25 and clinic B had the second lowest R^2 at 0.28. The following table shows both Spearman’s correlation coefficients and the multiple correlation coefficients from linear regression models within each of the 11 clinics that served as test sets in the LOCO-CV.

Clinic	R ²	R ² [log(Y + 1)]	Spearman's Cor.
A	0.56	0.36	0.73
B	0.28	0.31	0.44
C	0.45	0.35	0.69
D	0.25	0.16	0.40
E	0.5	0.52	0.70
F	0.39	0.34	0.58
G	0.59	0.5	0.72
H	0.49	0.47	0.66
I	0.34	0.23	0.53
J	0.31	0.33	0.46
K	0.45	0.36	0.63

Tabel R1: *The Spearman's correlation coefficient is between the number of MII oocytes collected and the total follicle size count on the trigger day in the test sets. R² scores are from two different regression models on the test sets where the response is the number of MII oocytes collected and the explanatory variables are individual follicle size counts on the day of trigger. The first model is with the outcome data as is, and the second with log(Y + 1) as the response variable, where Y is the number of MII oocytes collected.*

Overall, it seems that low R² scores are reflective of the relatively low correlation between the number of MII oocytes and follicle size counts in certain clinics. This is therefore likely due to other factors such as inter-operator variability and ultrasound measurement error (Liang et al, 2020). Although there may be other biological factors that could explain the variation in the number of MII oocytes, most of the variation is likely due to individual follicle size counts, as discussed in lines 99-103 of the manuscript, where we show that adding further variables of interest only marginally improved the prediction error.

Furthermore, as the reviewer has rightly pointed out, the R^2 score relates to the mean MII oocytes in the test sets. It is also more sensitive to outlying large errors than MAE due to the squaring of the errors. We have expanded the results of the MII oocyte model to show the predictive performance metrics for each clinic fold in **Figure R3**, as well as providing data summaries for MII oocytes in each training/test fold:

Figure R3: Graphical representation of the variability in prediction performance for the MII Oocyte model ($n=14,140$ patients) in each of the clinics labeled A-K.

To present this information to the reader, we have updated **Supplementary Table 1** to provide more granular detail on each clinic including the predictive performance, the minimum, maximum, mean, standard deviation, median, and the interquartile range, for MII and all oocytes, per clinic.

Clinic	Outcome	MAE	R2	MedAE	Min	LQ	Median	Mean	UQ	Max	IQR	SD
A	All oocytes	3.84	0.60	2.61	1.00	8.00	13.00	14.32	19.00	76.00	11.00	8.93
	MII oocytes	3.47	0.50	2.26	0.00	6.00	9.00	10.87	15.00	54.00	9.00	7.26
B	All oocytes	3.64	0.23	2.82	1.00	7.00	11.00	11.14	15.00	36.00	8.00	5.56
	MII oocytes	3.16	0.12	2.61	0.00	4.00	7.00	7.35	10.00	28.00	6.00	4.30
C	All oocytes	4.08	0.54	2.70	1.00	6.00	10.00	12.78	17.25	71.00	11.25	9.37
	MII oocytes	3.76	0.45	2.50	0.00	4.00	8.00	9.87	13.00	58.00	9.00	7.80
D	All oocytes	4.37	0.18	3.02	1.00	7.00	11.00	11.80	16.00	45.00	9.00	6.79
	MII oocytes	4.53	0.15	3.39	0.00	4.00	8.00	8.50	12.00	41.00	8.00	6.64
E	All oocytes	4.10	0.41	2.56	1.00	7.00	11.00	12.62	16.00	70.00	9.00	8.45
	MII oocytes	3.58	0.34	2.27	0.00	5.00	8.00	10.11	14.00	68.00	9.00	6.94
F	All oocytes	3.88	0.49	2.73	1.00	7.00	11.00	12.47	16.00	74.00	9.00	8.11
	MII oocytes	3.31	0.38	2.28	0.00	5.00	8.00	8.74	11.00	33.00	6.00	5.78
G	All oocytes	3.67	0.59	2.58	0.00	7.00	11.00	12.81	17.00	70.00	10.00	8.04
	MII oocytes	3.33	0.53	2.38	0.00	5.00	9.00	10.10	13.00	54.00	8.00	6.68
H	All oocytes	3.94	0.51	2.70	1.00	8.00	12.00	13.92	18.00	86.00	10.00	8.23
	MII oocytes	3.66	0.42	2.44	0.00	6.00	10.00	11.38	15.00	65.00	9.00	6.95
I	All oocytes	3.28	0.48	2.41	0.00	6.00	9.00	10.12	13.00	42.00	7.00	6.19
	MII oocytes	3.59	0.29	2.62	0.00	3.00	6.00	7.40	10.00	37.00	7.00	5.77
J	All oocytes	3.72	0.36	2.68	1.00	7.00	11.00	12.36	16.00	35.00	9.00	6.31
	MII oocytes	3.32	0.26	2.57	0.00	5.00	8.00	8.86	12.00	30.00	7.00	5.17
K	All oocytes	3.83	0.53	2.72	1.00	9.00	13.00	14.27	19.00	62.00	10.00	7.86
	MII oocytes	3.52	0.44	2.61	0.00	6.00	10.00	10.91	14.50	48.00	8.50	6.57
All	All oocytes	3.85	0.45	2.68	0.00	7.00	11.00	12.54	16.00	86.00	9.00	7.78
	MII oocytes	3.60	0.35	2.59	0.00	5.00	8.00	9.62	13.00	68.00	8.00	6.59

Supplementary Table 1: Model performance metrics per clinic for all oocytes (n=19,082) and metaphase-II (MII) oocytes (n=14,140) as outcome variables. For each model, the mean absolute error (MAE), coefficient of determination (R2), and median absolute error (MedAE) are reported per clinic when serving as the test set. Similarly, for the outcome variables all oocytes and MII oocytes, the minimum, lower quartile (LQ), median, mean, upper quartile (UQ), maximum, interquartile range (IQR), and standard deviation (SD) measures are reported. The final row represents data summaries for the whole dataset.

From **Figure R3** and **Supplementary Table 1** above, it can be seen that certain clinics have lower R² than others, such as clinic D (0.15) and clinic B (0.12), especially compared with clinic A (0.50). Due to averaging for the final results of the LOCO-CV, these reduced the mean R² from 0.40 to 0.35.

We have added the following sentence to the results:

“Performance of the models predicting all oocytes and MII oocytes are presented for each individual clinic in Supplementary Table 1.”

We hope that the further analyses and granular detail provided here provide sufficient insight into explaining model performance and reliability.

References:

Steyerberg, E.W. and Harrell Jr, F.E., 2016. Prediction models need appropriate internal, internal-external, and external validation. *Journal of clinical epidemiology*, 69, p.245.

Collins, G.S., Dhiman, P., Ma, J., Schlüssel, M.M., Archer, L., Van Calster, B., Harrell, F.E., Martin, G.P., Moons, K.G., Van Smeden, M. and Sperrin, M., 2024. Evaluation of clinical prediction models (part 1): from development to external validation. *bmj*, 384.

Liang, X., Fang, J., Li, H., Yang, X., Ni, D., Zeng, F. and Chen, Z., 2020. CR-Unet-based ultrasonic follicle monitoring to reduce diameter variability and generate area automatically as a novel biomarker for follicular maturity. *Ultrasound in Medicine & Biology*, 46(11), pp.3125-3134.

Reuvenny, S., Youngster, M., Luz, A., Hourvitz, R., Maman, E., Baum, M. and Hourvitz, A., 2024. An artificial intelligence-based approach for selecting the optimal day for triggering in antagonist protocol cycles. *Reproductive BioMedicine Online*, 48(1), p.103423.

Fanton, M., Nutting, V., Solano, F., Maeder-York, P., Hariton, E., Barash, O., Weckstein, L., Sakkas, D., Copperman, A.B. and Loewke, K., 2022. An interpretable machine learning model for predicting the optimal day of trigger during ovarian stimulation. *Fertility and Sterility*, 118(1), pp.101-108.

• It would also be very helpful to include a plot of the predicted vs. known values (for each clinic or all of them together). This visualization will provide a more intuitive understanding of the model's prediction performance and reliability.

We agree with the reviewer that a visualization of the predicted vs known values would be helpful in understanding the performance of the model. We have now included a plot of predicted versus actual values with data-points colored by clinic as **Supplementary Figure 1**.

Supplementary Figure 1: *Predicted versus actual values for the metaphase-II (MII) oocyte model (n=14,140 patients) for all clinics A-K.*

We have added this **Supplementary Figure 1** and the following sentence to the manuscript on lines 80-81:

“... and a plot of predicted versus actual MII oocytes collected color-coded by each clinic is presented in Supplementary Figure 1.”

2. Statistics on the Data:

• **Comprehensive statistics on the dataset are essential for assessing the context and reliability of the results. The authors should provide detailed information, including the mean and standard deviation (STD), as well as the range (minimum and maximum values) for all variables presented in Table 2.**

These additional details will greatly enhance the clarity of the relatively low performance model presented in the paper.

We thank the reviewer for highlighting the need for comprehensive statistics to assess the results reliably and within relevant context. **Table 2** has now been expanded to include: minimum, lower quartile, median, mean, upper quartile, maximum, interquartile range, standard deviation, N number, and percentage missing values.

Variable	Min	LQ	Median	Mean	UQ	Max	IQR	SD	N	Missing
Age at treatment	18.20	31.60	34.70	34.61	38.00	49.00	6.40	4.50	19,080	<1%
Body mass index	19.00	21.72	24.17	24.98	27.51	47.25	5.79	4.25	10,965	43%
Antral follicle count	1.00	8.00	15.00	17.39	23.00	130.00	15.00	13.28	9,620	50%
Days to trigger	4.00	10.00	11.00	11.34	12.00	32.00	2.00	1.99	19,072	<1%
No. follicles on DoT	3.00	9.00	14.00	16.16	20.00	90.00	11.00	9.86	19,082	0%
No. oocytes collected	0.00	7.00	11.00	12.54	16.00	86.00	9.00	7.78	19,082	0%
No. MII oocytes	0.00	5.00	8.00	9.62	13.00	68.00	8.00	6.59	14,140	26%
No. 2PN zygotes	0.00	3.00	6.00	7.15	10.00	58.00	7.00	5.18	17,822	7%
No. HQ blastocysts	0.00	1.00	2.00	3.17	4.00	31.00	3.00	3.05	17,488	8%
Mature oocyte yield	0.00	0.40	0.59	0.66	0.83	5.86	0.43	0.41	14,140	26%
Maturation rate	0.00	0.67	0.80	0.76	0.92	1.00	0.26	0.23	14,140	26%
Fertilization rate	0.00	0.55	0.71	0.69	0.86	1.00	0.31	0.22	12,757	33%
Blastulation rate	0.00	0.29	0.50	0.48	0.67	1.00	0.38	0.29	16,484	14%
Live birth rate	-	-	-	30.48%	-	-	-	-	12,724	33%

Table 2. Patient demographics and treatment cycle information from the total population of 19,082 participants. **Data are reported as the minimum, lower quartile (LQ), median, mean, upper quartile (UQ), maximum, standard deviation (SD), and interquartile range (IQR).** Mature oocyte yield is defined as the number of metaphase-II (MII) mature oocytes collected **divided by** the total follicle count on the day of trigger (DoT) administration. No.: number of; 2PN: two-pronuclear; HQ: high-quality.